# BDNF signaling in correlation-dependent structural plasticity in the developing visual system

Elena Kutsarova[1,2], Anne Schohl[1], Martin Munz[1,3,4], Alex Wang[1,5], Yuan Yuan Zhang[1,6], Olesia M. Bilash[1,7], Edward S. Ruthazer[1]*

1 Montreal Neurological Institute-Hospital, McGill University, Montreal, Canada, 2 Max Planck Institute for Brain Research, Frankfurt, Germany, 3 Institute of Molecular and Clinical Ophthalmology Basel, Basel, Switzerland, 4 Department of Ophthalmology, University of Basel, Basel, Switzerland, 5 Interdepartmental Neuroscience, Yale University, New Haven, Connecticut, United States of America, 6 Faculty of Medicine, University of Ottawa, Ottawa, Canada, 7 NYU Neuroscience Institute, New York University, New York, New York, United States of America

* edward.ruthazer@mcgill.ca

**Data Availability Statement:** Raw axon reconstructions from which the analyses were performed are available at https://figshare.com/projects/BDNF_signaling_in_correlation-

## Abstract

During development, patterned neural activity instructs topographic map refinement. Axons with similar patterns of neural activity converge onto target neurons and stabilize their synapses with these postsynaptic partners, restricting exploratory branch elaboration (Hebbian structural plasticity). On the other hand, non-correlated firing in inputs leads to synapse weakening and increased exploratory growth of axons (Stentian structural plasticity). We used visual stimulation to control the correlation structure of neural activity in a few ipsilaterally projecting (ipsi) retinal ganglion cell (RGC) axons with respect to the majority contralateral eye inputs in the optic tectum of albino *Xenopus laevis* tadpoles. Multiphoton live imaging of ipsi axons, combined with specific targeted disruptions of brain-derived neurotrophic factor (BDNF) signaling, revealed that both presynaptic p75[NTR] and TrkB are required for Stentian axonal branch addition, whereas presumptive postsynaptic BDNF signaling is necessary for Hebbian axon stabilization. Additionally, we found that BDNF signaling mediates local suppression of branch elimination in response to correlated firing of inputs. Daily in vivo imaging of contralateral RGC axons demonstrated that p75[NTR] knockdown reduces axon branch elongation and arbor spanning field volume.

## Introduction

Sensory experience during early development is crucial for the formation of precise topographic maps throughout the brain [1]. Visual stimulation drives action potential firing in retinal ganglion cells (RGCs) of fish and amphibians during the period when they first extend into the optic tectum, whereas in mammalian development, spontaneous waves of retinal activity drive RGC patterned activity [2–5]. Therefore, temporal correlation in firing patterns of RGCs is indicative of their mutual proximity in the retina, and the developing visual system uses this information to instruct structural and functional refinement of retinotopic and higher-order

dependent_structural_plasticity_in_the_
developing_visual_system/160870 Custom
MATLAB software (CANDLE denoising) used in this
study is available for free download at https://sites.
google.com/site/pierrickcoupe/softwares/
denoising/multiphoton-image-filtering Custom
arbor morphology analysis code is available at
https://github.com/ekutsarova/Dynamic-arbor-
morphology-analysis.

**Funding:** This work was supported by Canadian
Institutes of Health Research Foundation Grant
(FDN-143238 to ESR) and the following
studentships: Jeanne Timmins Costello and
Molson Neuroengineering Studentships (EK), Ann
and Richard Sievers Award in Neuroscience (MM),
McGill Summer Undergraduate Research Award
(AW) and a Natural Science and Engineering
Research Council CREATE Neuroengineering
Training Grant Summer Research Award (YYZ).
The funders had no role in study design, data
collection and analysis, decision to publish, or
preparation of the manuscript.

**Competing interests:** The authors have declared
that no competing interests exist.

**Abbreviations:** BDNF, brain-derived neurotrophic
factor; DAS, Darkness-Asynchronous-
Synchronous; NMDAR, N-methyl-D-aspartate
receptor; MO, morpholino oligonucleotide; NMJ,
neuromuscular junction; RGC, retinal ganglion cell.

feature maps [6–8]. Concretely, co-activation of RGC inputs, read out by calcium flux through postsynaptic N-methyl-D-aspartate receptors (NMDARs), stabilizes retinotectal synapses and their axonal arbors [7,9–14]. This correlation-dependent structural and functional stabilization is referred to as "Hebbian plasticity" [15]. In contrast, asynchronous firing of RGC axons with respect to their neighboring inputs leads to synaptic weakening, accompanied by an increase in axonal branch dynamics and exploratory growth, referred to as "Stentian plasticity" [7,16,17]. Previous in vivo imaging and electrophysiological recordings indicate that activity in neighboring axons promotes the Stentian synaptic weakening and branch elaboration of those axons that have not otherwise been stabilized by correlated firing and NMDAR activation [7,17].

Brain-derived neurotrophic factor (BDNF) has been implicated as a potent modulator of synaptic and structural plasticity throughout the brain [18]. BDNF can be synthesized and released in a constitutive or activity-dependent manner from axonal terminals and dendrites [19–24]. It can be released as an active precursor form, proBDNF, which exerts its function mainly via $p75^{NTR}$ and sortilin signaling, or as the cleaved mature form (mBDNF), which signals mainly via TrkB and $p75^{NTR}$/TrkB [25–28]. Furthermore, conversion of proBDNF to mBDNF is critical for correlation-dependent synaptic strengthening [29,30]. The specifics of BDNF synthesis, secretion, and receptor signaling create multilayered regulation underlying the diversity of functions served by BDNF in the brain. For example, at the developing neuromuscular synapses, proBDNF signaling through presynaptic $p75^{NTR}$ is required for axonal retraction, whereas mBDNF signaling through presynaptic TrkB leads to axonal stabilization [31,32]. In the developing retinotectal system, exogenous application of BDNF induces RGC axon branching and growth, whereas depletion of endogenous BDNF reduces presynaptic synaptobrevin punctum formation and axonal branch stabilization [33–35]. BDNF signaling through the TrkB receptor is necessary for Hebbian synaptic strengthening induced by visual conditioning in the *Xenopus* retinotectal system, whereas proBNDF signaling through $p75^{NTR}$ facilitates synaptic weakening [12,36]. Overall, the literature to date suggests that BDNF could both be involved in increased exploratory growth (Stentian plasticity), as well as in structural and synaptic stabilization (Hebbian plasticity) in the optic tectum in *Xenopus*.

In this study, we set out to test what aspects of BDNF signaling are respectively involved in Hebbian and Stentian structural plasticity. We took advantage of the existence of ectopic ipsilaterally projecting (ipsi) RGC axons in albino *Xenopus laevis* tadpoles, which typically consist of just 1 or 2 axons and which spontaneously occur in fewer than half of all albino and wild-type animals [7,37,38]. Since these ipsi RGC axons are not observed in all tadpoles, it is more likely that they are a result of rare errors in pathfinding, rather than constituting a functionally specialized class of RGCs. Exploiting the presence of an ipsi axon, we used optical fibers to present light to each eye, visually stimulating the ipsi axon either in or out of synchrony with the contralateral eye from which the bulk of retinotectal input originates. Furthermore, we selectively knocked down either $p75^{NTR}$ or TrkB in the ipsi axons, providing evidence by in vivo imaging for the presynaptic involvement of $p75^{NTR}$, and TrkB, in Stentian axonal branching and growth driven by uncorrelated activity. We further showed that Hebbian stabilization in response to correlated firing was blocked by application of an extracellular TrkB-Fc to prevent BDNF signaling. Daily imaging of contralaterally projecting RGC axons revealed that the $p75^{NTR}$ receptor is required for axonal arbor growth and 3D expansion, in line with a role in Stentian plasticity. Our findings suggest that these opposing forms of plasticity both depend on neurotrophin signaling but have distinct sites of action and engage different receptors.

## Results

### Stentian and Hebbian structural plasticity are mediated by distinct components of BDNF signaling

We performed in vivo 2-photon imaging of EGFP-expressing ipsi RGC axonal arbors in the optic tectum every 10 min for up to 5 h while subjecting the animals to a Darkness-Asynchronous-Synchronous (DAS) visual stimulation protocol based on Munz and colleagues (2014) (**Fig 1A and 1B**). Ipsi RGC axons exhibit greater branch dynamics and growth when stimulated asynchronously with respect to the majority contra RGC inputs, but synchronous activation of the ipsi and contra axons results in arbor stabilization [7]. To examine the contributions of BDNF signaling, we employed 3 strategies: either intraventricular injection of TrkB-Fc to sequester released BDNF acutely, or co-electroporation of EGFP and antisense morpholino oligonucleotide (MO) against p75$^{NTR}$ (p75-MO) or TrkB (TrkB-MO) in the eye to achieve presynaptic knockdown of these receptors in the RGCs (**Figs 1A and S1**). MOs were labeled with the red fluorescent dye lissamine that filled the RGC out to its axonal terminal, permitting knockdown to be confirmed by the presence of both EGFP and lissamine fluorescence in individual ipsi axons. We quantified changes in the rates of branch additions and losses per axon for each 10 min imaging interval throughout the DAS protocol (**Fig 1B and 1C**). Importantly, in the optic tectum ipsilateral to the electroporated eye, only the ipsi RGC axon contains the MO, whereas the majority RGC inputs, originating from the contralateral eye, are unmanipulated.

Ipsi axons containing Ctrl-MO (Control) exhibited a Stentian increase in branch dynamics during asynchronous stimulation of the 2 eyes, compared to Hebbian stabilization induced by synchronous stimulation (**Fig 1D–1G**), consistent with our previous report (**S2 Fig**) [7]. Our previous study showed the strongest increase in branch additions within the first hour of asynchronous stimulation, whereas the strongest increase in branch loss occurred during the second hour of asynchronous stimulation. Therefore, we focused on these 2 time periods. The higher rate of branch addition during asynchronous stimulation compared to synchronous stimulation was prevented in p75-MO axons (**Fig 1D and 1F**), suggesting a direct role of the low-affinity neurotrophin receptor in Stentian plasticity. TrkB-MO in the ipsi axon also prevented the increase in branch addition during asynchronous stimulation (**Fig 1D and 1F**). However, the persistence of a relative difference in TrkB-MO axon branch additions between asynchronous and synchronous stimulation conditions suggests that although blocking axonal TrkB signaling may reduce activity-dependent branching, it does not fully eliminate the effects of correlated activity (**S3A and S3C Fig**).

On the other hand, depletion of BDNF signaling by extracellular application of TrkB-Fc prevented the Hebbian stabilization of branch dynamics, which manifests as a suppression of new branch additions and losses during synchronous binocular stimulation (**Fig 1D–1G**). A significant increase in branch additions during synchronous stimulation was observed exclusively in TrkB-Fc treated, but not in neurotrophin receptor knockdown RGC axons, implying that BDNF must act on targets other than the axon itself, presumably the postsynaptic cells where it has been shown to mediate synaptic plasticity [36], to induce Hebbian retrograde signals that inhibit formation of new axonal branches.

### BDNF signaling helps suppress branch loss during synchronous stimulation

In the optic tectum, RGC axons continuously extend and retract processes in the neuropil during structural refinement of the developing retinotopic map. Axonal branch elimination occurs in parts of the arbor where synapses have failed to undergo stabilization, permitting

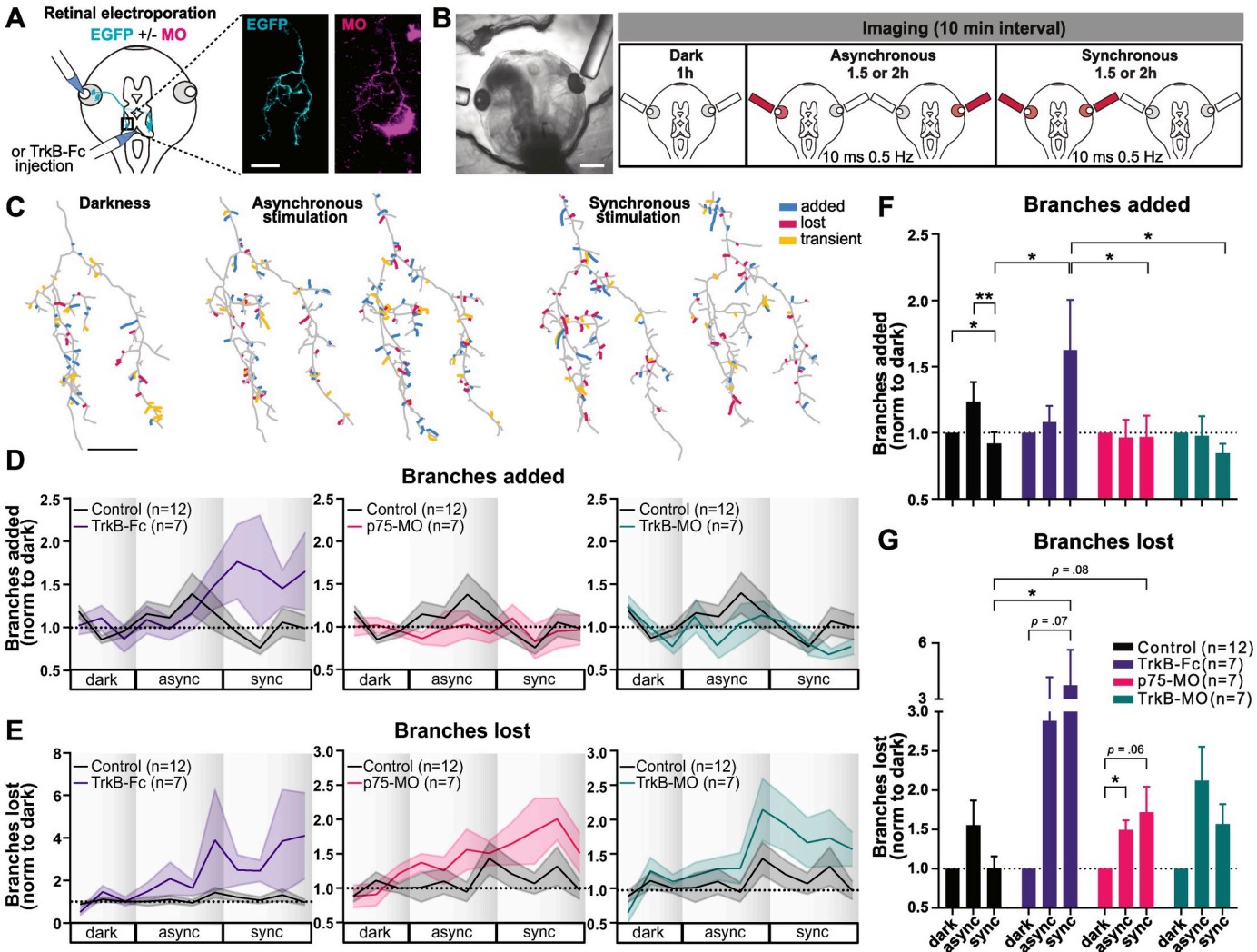

**Fig 1. Blocking distinct components of BDNF signaling differentially affects correlation-dependent branch dynamics in ipsilaterally projecting RGC axons.** (A) Retinal co-electroporation of EGFP and MO to visualize ipsi axons with receptor knockdown. Two-photon z-series projection of ipsi axon with EGFP (cyan) and lissamine-tagged Ctrl-MO (magenta). In magenta, 2 microglia are also visible. For TrkB-Fc experiments, intraventricular injection was performed >1 h prior to imaging. (B) Visual stimulation (10 ms light flash; 0.5 Hz) delivered via optic fibers positioned by the tadpoles' eyes. Ipsi axons were imaged every 10 min: 1 h Darkness, 1.5 h or 2 h Asynchronous, 1.5 h or 2 h Synchronous stimulation (DAS: shorter protocol used for intraventricular TrkB-Fc). (C) Reconstructed arbors showing added, lost, and transient branches (added and lost within the summarized period). Drawings summarize branch dynamics over 1 h. Time course (20 min average) of branch (D) additions and (E) losses normalized to the average in darkness, DAS (1 h, 1.5 h, 1.5 h). Average branch (F) addition in the first and (G) loss rates in the second half of stimulation. Significant interactions in the two-way mixed design model: (F) $p = 0.0189$; (G) $p < 0.001$. (Within-subject factor: stimulation; between-subject factor: BDNF manipulation); post hoc tests corrected for multiple comparisons (BKY two-stage linear step-up procedure): $^*p \leq 0.05$, $^{**}p \leq 0.01$. Graphs show mean ± SEM. (D–G) Control ($n = 12$), TrkB-Fc ($n = 7$), p75-MO ($n = 7$), TrkB-MO ($n = 7$). Scale bars: (A, C) 20 μm; (B) 500 μm. The data used to generate Fig 1D and 1E can be found in S1 Data and Fig 1F and 1G in S2 Data. BDNF, brain-derived neurotrophic factor; DAS, Darkness-Asynchronous-Synchronous; EGFP, enhanced green fluorescent protein; MO, morpholino oligonucleotide; RGC, retinal ganglion cell.

axon pruning and topographic map refinement [8]. TrkB-Fc depletion of BDNF signaling led to an increase in the rates of axonal branch loss in response to synchronous stimulation compared to control ipsi axons, with even more branch elimination during synchronous than asynchronous stimulation (**Figs 1E, 1G,** S3B **and** S3D). This effect on branch loss during synchronous stimulation was at least partially replicated by p75 knockdown in the ipsi axon. On the other hand, branch elimination rates were not elevated during synchronous compared to

asynchronous stimulation in ipsi axons with TrkB-MO. These data show that BDNF release, likely involving the activation of presynaptic p75$^{NTR}$, contributes to the Hebbian suppression of branch loss that occurs in response to correlated firing.

Taken together, these findings strongly implicate BDNF release in the Hebbian suppression of branch addition and elimination that occurs in response to synchronous firing of inputs. Moreover, presynaptic p75$^{NTR}$ appears to be required for the differential response of the axon to asynchronous and synchronous activity, as the Stentian addition of new branches during asynchronous stimulation and the Hebbian suppression of branch elimination during synchronous firing were both lost in p75-MO cells, which responded similarly to the 2 stimulation conditions. On the other hand, presynaptic knockdown of TrkB did not eliminate differences in branch dynamic behaviors between synchronous versus asynchronous conditions.

## BDNF signaling spatially restricts branch elimination events

Local action of BDNF has previously been shown to alter synaptic maturation and dendritic morphology in cells in close proximity to release sites [39–42]. We sought to further explore the role of BDNF signaling in the spatial organization of axonal branch addition and elimination events in response to patterned activity (Fig 2A and 2B). To assess whether remodeling events were spatially clustered, indicative of local signaling on the arbor, we extracted the pairwise distances for all addition and for all elimination events during darkness, asynchronous, and synchronous stimulation (Fig 2C) and calculated a mean event pair distance for each stimulation period (Figs 2D, S4C and S4D), shown for the example axon (Fig 2C). Differences in arbor size and shape may impact measurements of mean event pair-distance. We therefore performed Monte Carlo simulations of the locations of events on the arbor (S4A and S4B Fig) and used the simulated mean event pair-distances to correct for differences in arbor morphology for all analyses. The corrected mean pairwise distance between elimination events was significantly lower during synchronous stimulation compared to darkness in control axons, due to fewer branch elimination events occurring far apart from each other during correlated firing induced by synchronous stimulation (Fig 2D and 2E). These data reveal that correlated activity causes branch eliminations to become restricted to a relatively smaller portion of the total arbor. Extracellular depletion of BDNF with TrkB-Fc and, to a much lesser extent, knockdown of the receptors in RGC axons, resulted in a redistribution of branch elimination events, no longer favoring the event proximity normally seen with synchronous stimulation (Fig 2E). In particular, there was a significant difference between pair distances of branch loss events during synchronous stimulation in TrkB-Fc and control animals. Interestingly, this spatial phenomenon was limited to branch elimination, as we observed no difference in mean pairwise distances of addition events in control axons across the stimulation periods (Fig 2F). These findings point to a model in which secreted neurotrophin can influence branch eliminations and stabilization within spatially constrained local zones of action. This suggests that the suppression of branch loss during synchronous activation, mediated by neurotrophin release (Fig 1G), acts within a spatially restricted part of the arbor, creating zones where branches are relatively protected from elimination (Fig 2B and 2D).

## Axon arbor elaboration over days relies on p75$^{NTR}$

Overexpression of the truncated TrkB isoform, which blocks BDNF signaling by sequestering BDNF or forming nonfunctional TrkB dimers, has been shown to reduce RGC axonal arborization over a day [43]. We therefore set out to identify the roles of both p75$^{NTR}$ and TrkB in long-term axonal arbor elaboration by performing knockdown of each of the receptors under conditions of normal visual experience. In this case, we performed daily imaging over 4 days of contralaterally projecting RGC axons co-electroporated with EGFP and p75-MO or

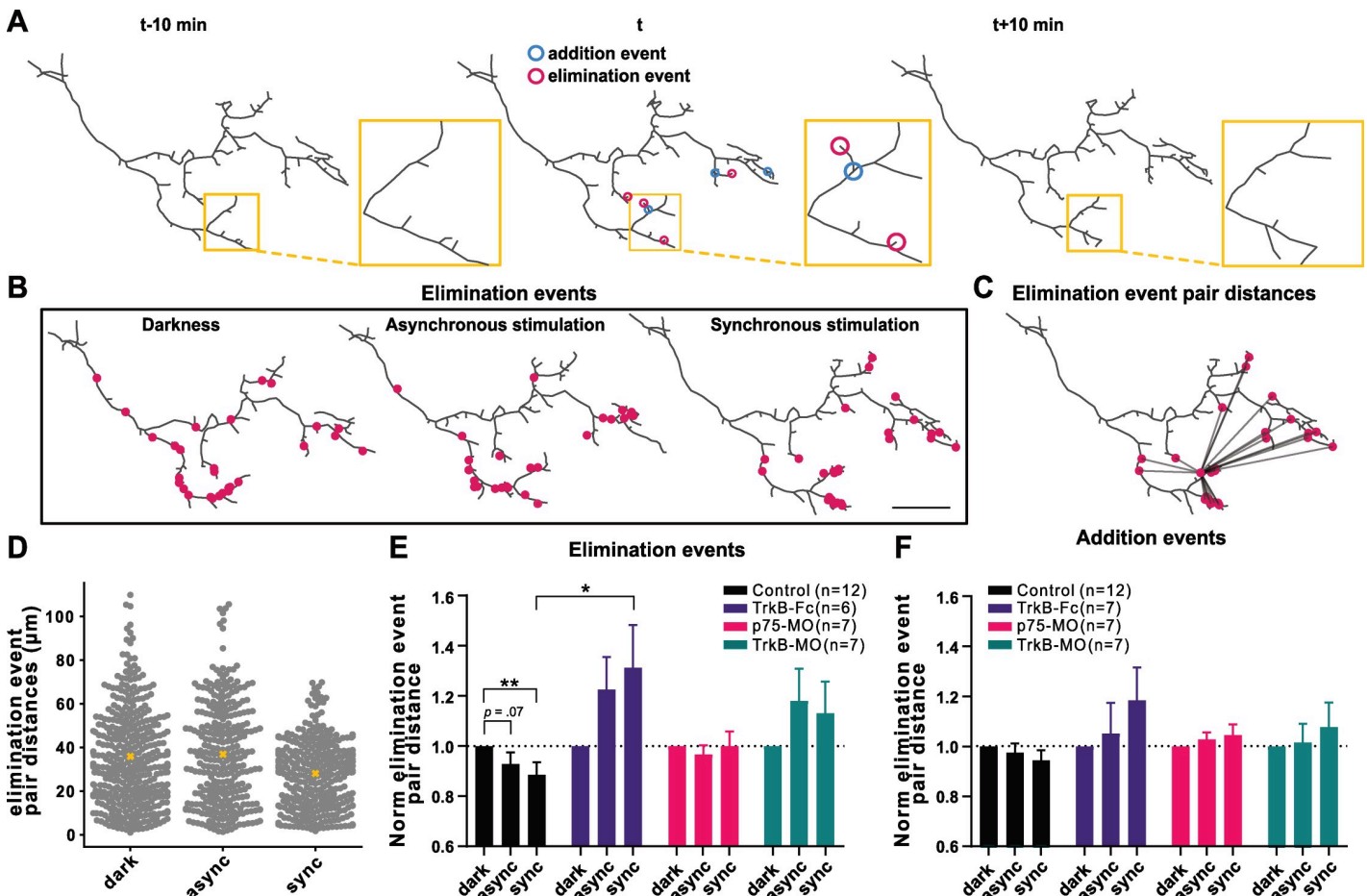

**Fig 2. Role of BDNF signaling in local branch elimination induced by synchronous stimulation in ipsilaterally projecting RGC axons.** (A) Three consecutive reconstructed arbors from a control ipsi axon during synchronous stimulation. Addition and elimination event locations (for time point "*t*") are defined as the coordinates of the branch point of the newly added axonal branch ($\geq$1.5 μm) between *t-10 min* and *t* (blue), or terminal point of a lost branch between *t* and *t+10 min* (magenta), respectively. (B) All elimination events during DAS (1 h, 1.5 h, 1.5 h) are superimposed on the reconstructed arbor (last time point of each stimulation period). (C) Illustration of distances between an elimination event and all the other elimination events (pair distances). (D) All the elimination event pair distances for the axon from (B). Means are denoted as yellow crosses. Normalized branch (E) elimination and (F) addition event pair distances. To correct for changes in arbor size, each point represents the ratio of observed-to-randomized (S4 Fig) mean event pair distances, normalized to darkness for each axon. Interaction in the two-way mixed design model: (E) $p < 0.001$; (F) $p = 0.0562$. Post hoc tests corrected for multiple comparisons (BKY two-stage linear step-up procedure): *$p \leq 0.05$, **$p \leq 0.01$. (E, F) Data are shown as mean + SEM. (E, F) Control ($n = 12$), (E) TrkB-Fc ($n = 6$), (F) TrkB-Fc ($n = 7$), (E, F) p75-MO ($n = 7$), (E, F) TrkB-MO ($n = 7$). (B) Scale bar: 20 μm. The data used to generate Fig 2E and 2F can be found in S3 Data. BDNF, brain-derived neurotrophic factor; DAS, Darkness-Asynchronous-Synchronous; RGC, retinal ganglion cell.

TrkB-MO (**Fig 3A–3C**). RGC axons knocked-down for TrkB exhibited a significantly greater accumulation of branch tips compared to axons with p75-MO (**Fig 3D and 3E**). Furthermore, p75$^{NTR}$ knockdown resulted in a significant decrease in the skeletal length of the axon compared to control axons (electroporated with Ctrl-MO) and RGCs electroporated with TrkB-MO (**Fig 3F**). In addition, binning terminal segments (**Fig 3D**) by length revealed that as axonal arbors became more complex over 4 days, the proportion of terminal segments categorized as short (1 to 5 μm) decreased in control and TrkB-MO axons, whereas it remained unchanged in the p75-MO axons, consistent with their failure to undergo progressive elaboration (**Fig 3G**). These findings indicate that presynaptic p75$^{NTR}$ underlies new branch accumulation and the elongation of short filopodium-like branches over days, resulting in a more complex terminal arbor, whereas TrkB helps keep the arbor compact.

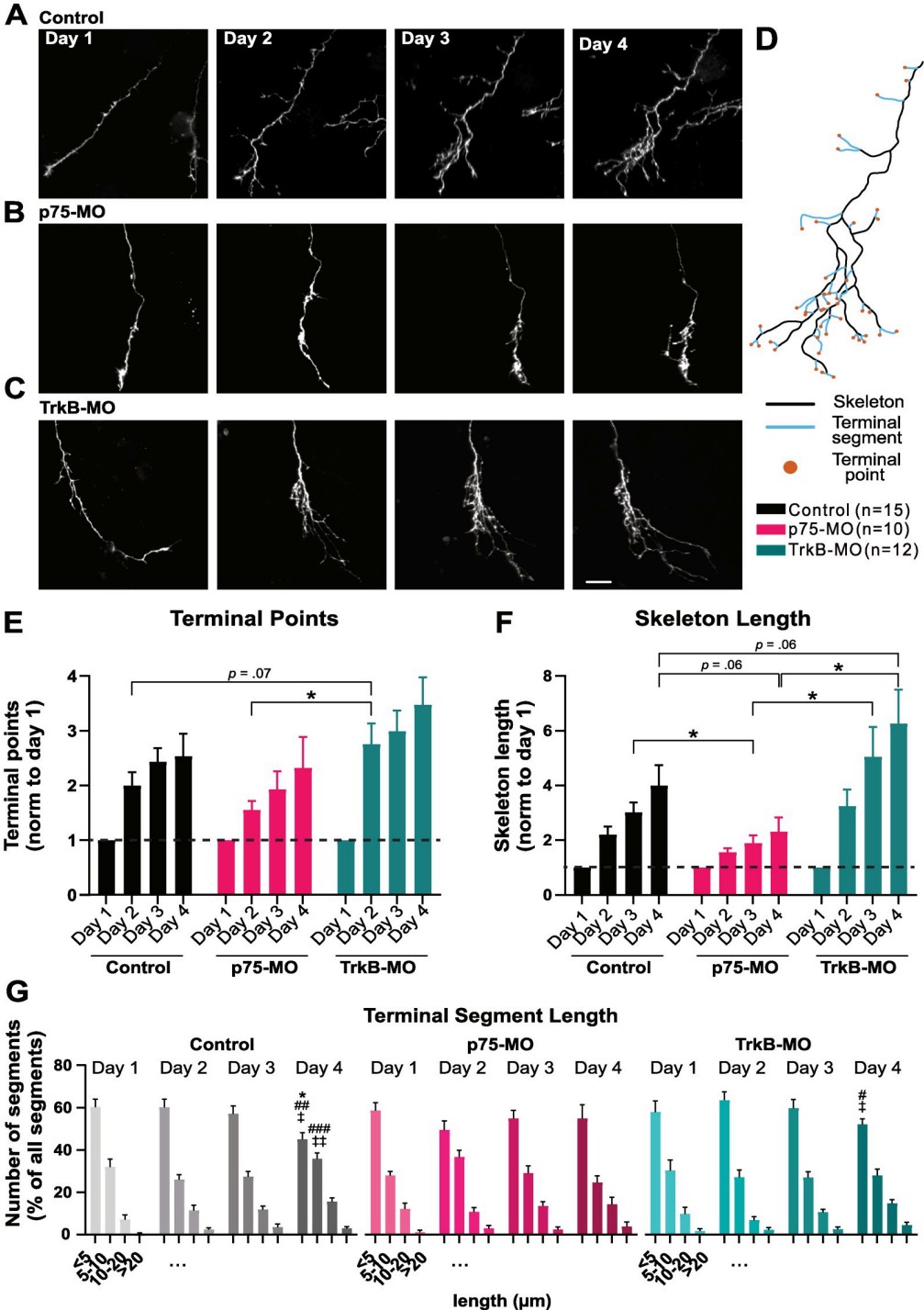

**Fig 3. Effects of retinal TrkB-MO and p75-MO on contralaterally projecting RGC axonal arbor elaboration over days.** (–C) Two-photon z-series projections over 4 days for axons co-electroporated with EGFP and (A) Ctrl-MO, (B) p75-MO, (C) TrkB-MO. (D) Reconstructed control arbor from day 4 showing axonal skeleton (black), terminal segments (blue), and terminal points (orange). (E, F) Morphometric analysis including (E) number of terminal points and (F) skeleton length, normalized to day 1. (G) Distribution of terminal segments (percent of total per axon) binned by length. Significant interactions in the two-way mixed design model: (E, F) $p < 0.001$; (G) $p = 0.04394$ ($<5$ μm bin); $p = 0.01981$ (5–10 μm bin). Post hoc tests corrected for multiple comparisons (BKY two-stage linear step-up procedure): (E, F) *$p \leq 0.05$, **$p \leq 0.01$, and (G) comparison between day 1 and day 4, (G) comparison between day 2 and day 4: #$p \leq 0.05$,

##$p \leq 0.01$, ###$p \leq 0.001$, (G) comparison between day 3 and day 4: ‡$p \leq 0.05$, ‡‡$p \leq 0.01$. Data represent (E–G) mean + SEM. (E–G) Control ($n = 15$), p75-MO ($n = 10$), TrkB-MO ($n = 12$). Scale bar: 20 μm applies to all images. The data used to generate Fig 3E and 3G can be found in S4 Data. MO, morpholino oligonucleotide; RGC, retinal ganglion cell.

## Roles of p75$^{\text{NTR}}$ and TrkB in RGC axonal arbor span enlargement

The retinotopic map occupies a 3D volume within the tectum [14], and therefore, a relevant metric of an axon's morphology and refinement is the volume of its arbor span. We calculated arbor span volume for contralaterally projecting RGC axons by obtaining a measure of the convexity of the 3D reconstructed axonal arbor, which was used to define an enclosing volume whose border lies between a tight fit and a convex hull around the arbor, as described previously [44] (**Fig 4A–4C**). We found that TrkB-MO arbors expand more rapidly over 4 days to occupy a greater volume in the optic tectum compared to p75-MO axons (**Fig 4D and 4E**). These observations suggest that p75$^{\text{NTR}}$ in the RGC may mediate enlargement of the axonal arbor span volume, whereas TrkB may contribute to maintaining the arbor volume compact.

## Discussion

### BDNF signaling in Hebbian suppression of axonal branch dynamics

Hebbian axonal branch stabilization induced by correlated firing is dependent on the activity of NMDARs [7,11]. Postsynaptic knockdown of NMDARs in optic tectal neurons results in increased rates of dynamic branch additions and retractions in RGC axons, suggesting the existence of a retrograde signal that promotes presynaptic axon arbor stabilization in response to NMDAR activation [13]. Activation of tectal NMDARs can induce the synaptic entry of calcium, leading to the activation of CaMKII in optic tectal neurons, which has been shown in live imaging experiments to mediate a retrograde modulation of presynaptic axonal dynamics and growth [45,46]. The identity of this putative retrograde signaling factor remains unclear, but several lines of evidence have suggested BDNF as a feasible candidate retrograde signal [47–50]. In our experiments, presynaptic p75$^{\text{NTR}}$ knockdown and sequestration of BDNF with TrkB-Fc both prevented the usual decrease in branch addition and loss caused by synchronous stimulation, suggesting that BDNF could act through activation of presynaptic p75$^{\text{NTR}}$ in Hebbian plasticity. However, the robust effects of sequestering extracellular BDNF were at best only partially replicated by axonal p75$^{\text{NTR}}$ knockdown (**Fig 1D and 1F**), suggesting that released BDNF likely also acts postsynaptically to drive synaptic changes that facilitate the release of additional retrograde signals (**Fig 5**). In support of this, in other systems, BDNF has been shown to be released downstream of postsynaptic NMDAR activation and to act as an autocrine signal mediating synaptic strengthening at dendritic spines [24,51]. This notion is supported in the retinotectal system by previous reports showing that BDNF downstream of NMDAR activation promotes the formation and stabilization of postsynaptic elements [35,36,52]. While a strength of our experimental design is the ability to knock down receptor in just the RGC axons while sparing postsynaptic partners in the tectum, an important caveat is that dendritic morphologies of electroporated RGCs in the retina are likely to be affected by receptor knockdown, as has been shown previously [53]. However, in that earlier study, it was reported that retinal BDNF manipulation did not affect RGC axonal morphology in the tectum.

### Presynaptic p75$^{\text{NTR}}$ and TrkB in Stentian exploratory growth

Knockdown of p75$^{\text{NTR}}$ and TrkB receptors prevented the Stentian increase in new branch addition that occurs when an axon does not fire in synchrony with its neighbors (**Fig 1D and 1F**) [7,17]. Furthermore, we found a decrease in branch tip accumulation over 4 days of

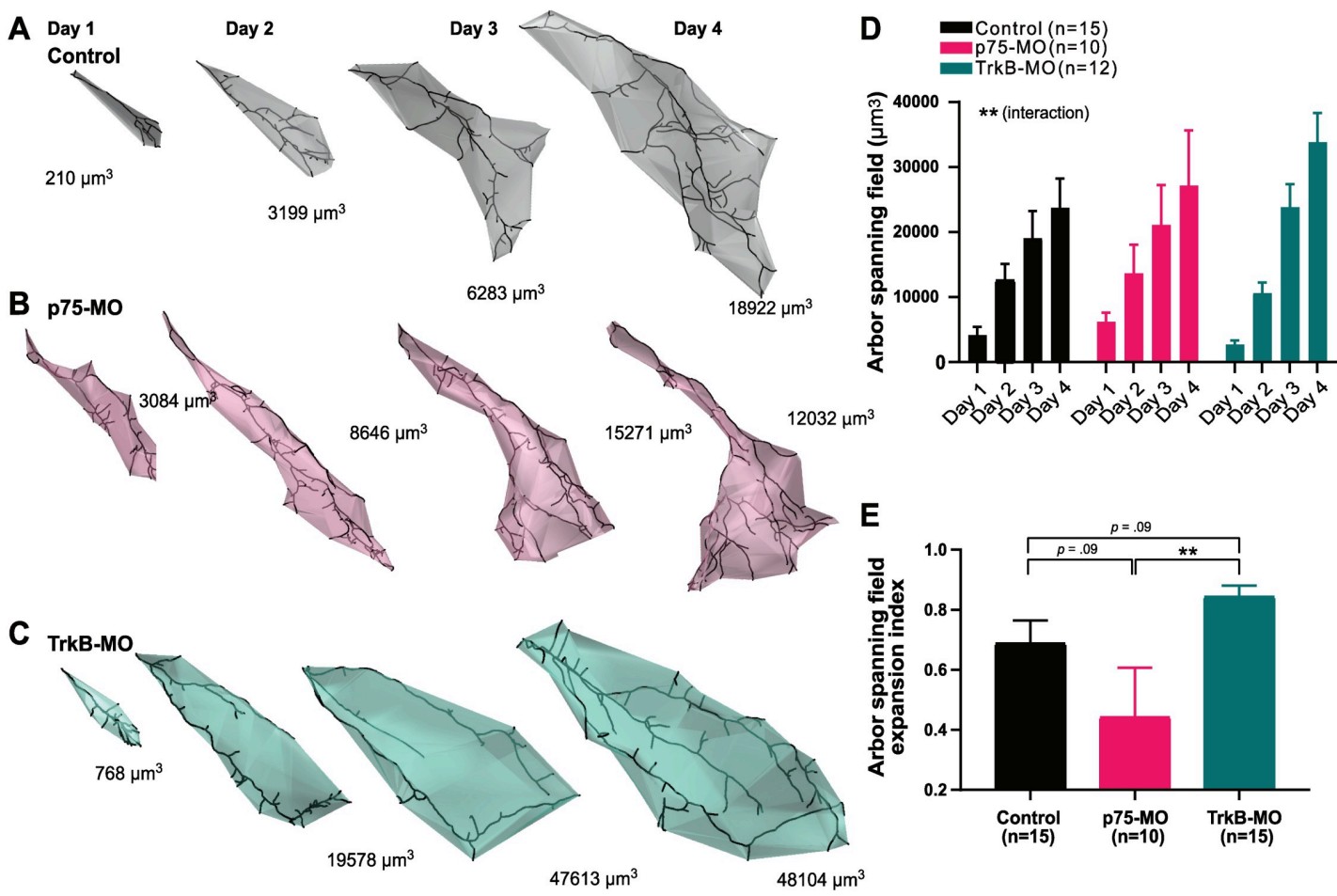

**Fig 4. Arbor span compactness of contralaterally projecting RGC axons is affected by retinal TrkB-MO and p75-MO.** (A–C) Three-dimensional spans of reconstructed RGC axon arbors over 4 days, co-electroporated with EGFP and MO: (A) Ctrl-MO, (B) p75-MO, and (C) TrkB-MO. (D) Significant interaction in the two-way mixed design model: $**p = 0.00119$. (E) TrkB-MO leads to a faster increase in arbor spanning field volume compared to p75-MO [Arbor spanning field expansion index = (day 4—day 1)/(day 1 + day 4)], analyzed by Kruskal–Wallis test, $p = 0.0175$, and pairwise post hoc tests corrected for multiple comparisons: $**p \leq 0.01$. Data represent mean + SEM. (D, E) Control ($n = 15$), p75-MO ($n = 10$), TrkB-MO ($n = 12$). The data used to generate Fig 4D can be found in S5 Data and Fig 4E in S6 Data. MO, morpholino oligonucleotide; RGC, retinal ganglion cell.

repeated imaging in p75-MO axons (**Fig 3E**). Previous work demonstrated that when inputs surrounding an axon actively fire while that axon is quiescent, Stentian "exploratory growth" of the inactive axon takes place, suggesting the possible release of an intercellular growth-promoting signal by neighboring cells in the optic tectum [17]. From our experiments, we can conclude that axonal p75$^{NTR}$ and possibly TrkB mediate the response to Stentian signals that promote branch addition and extension. TrkB may mediate a general increase in axonal branch addition during firing, as the difference between asynchronous and synchronous stimulation appears to be maintained in TrkB-MO, whereas p75$^{NTR}$ appears to underlie the differential responses to asynchronous and synchronous stimulation (**S3C Fig**). Putative p75$^{NTR}$ ligands such as proBDNF or other neurotrophins could be released directly by the neighboring axons, indirectly by the postsynaptic neurons or even by local glia (**Fig 5A and 5B**). Further investigation is required to reveal the identities and the exact sites of release of these potential ligands. ProBDNF-induced activation of p75$^{NTR}$ has been shown to promote synaptic

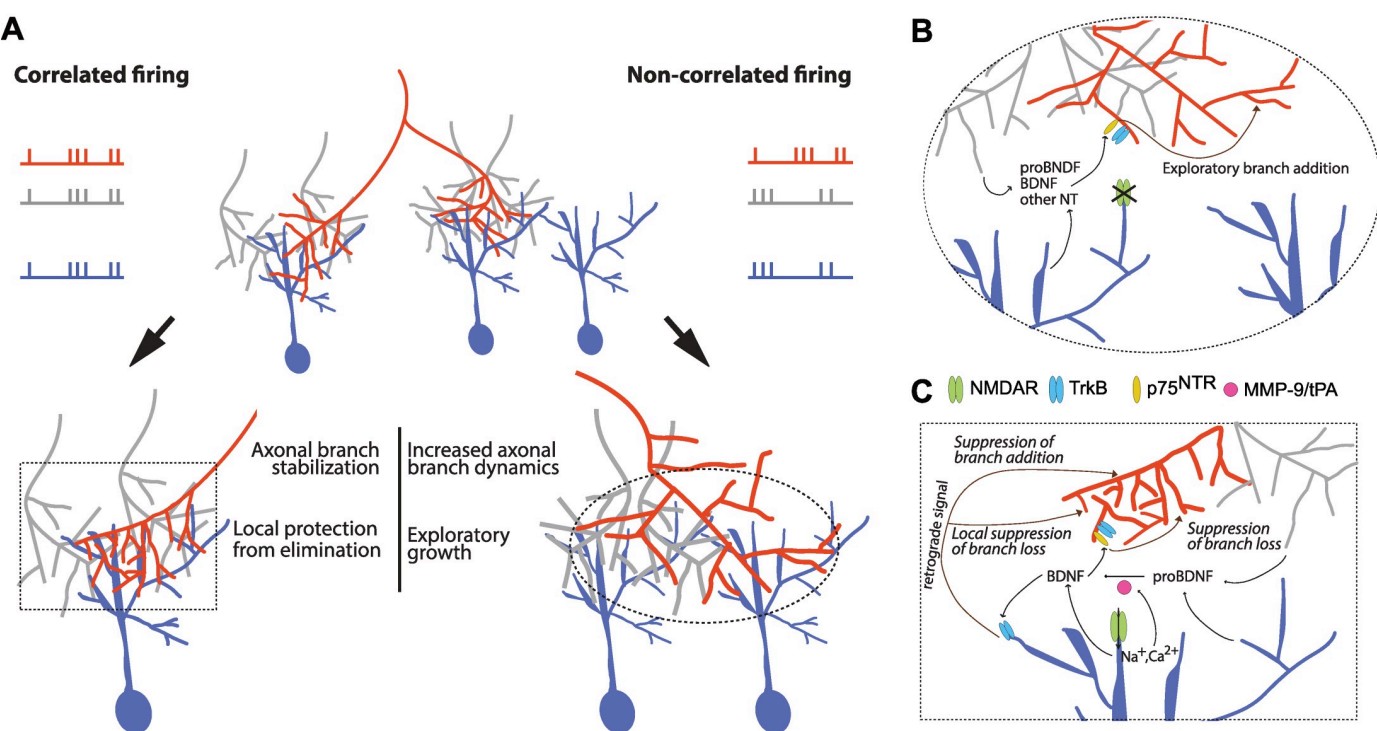

**Fig 5. Proposed model of BDNF signaling in correlation-dependent structural remodeling.** (A) Summary schematic of structural remodeling of an RGC axon of interest (red) instructed by patterned activity based on our current and previous studies: correlated firing (left) and non-correlated firing (right) of axon with neighboring inputs and postsynaptic partner. In conditions of poorly correlated firing, there is an increase in branch dynamics (addition and loss) and exploratory growth. Correlated firing results in a decrease in branch dynamics addition and loss, occurring within part of the axonal arbor. Protection against branch elimination occurs locally in parts of the arbor, strongly connected to tectal neurons (blue) via synapses in which postsynaptic NMDARs are strongly activated. Zoom-in of proposed molecular and cellular mechanisms underlying (B) Stentian and (C) Hebbian structural plasticity in the developing visual system. (B) Uncorrelated firing between an RGC axon (red) and its neighboring axons (gray), leads to a failure to activate postsynaptic NMDARs. A signal that promotes branch initiation and growth binds to p75$^{NTR}$ on the axon of interest (red) may be released by the neighboring axons (gray) or the postsynaptic partner (blue). We propose that this signal may be proBDNF. (C) Correlated firing of the RGC axon of interest (red) and its neighboring axons (gray) results in an activation of postsynaptic NMDARs, leading to an increase in the concentration of BDNF via release or cleavage of proBDNF (e.g., via MMP-9 or tPA). BDNF binds to postsynaptic TrkB that initiates a retrograde stabilization signal (of unknown identity) leading to suppression of branch addition and targeted decrease of branch loss. BDNF, brain-derived neurotrophic factor; MMP-9, Matrix metalloproteinase-9; NMDAR, N-methyl-D-aspartate receptor; NT, neurotrophin; RGC, retinal ganglion cell; tPA, tissue plasminogen activator.

weakening both in *Xenopus* tadpoles and in mice [27,29,36], consistent with our previous finding that exploratory axon growth is accompanied by a marked reduction in synaptic strength [7]. In addition, our results are in line with previous reports showing that injection of BDNF in the optic tectum drives a robust increase in RGC axonal branch number, an effect which could be the result of activation of either TrkB or p75$^{NTR}$ [33]. Previously, *Xenopus* RGC axons electroporated with TrkB.T1 have been shown to exhibit an increase in axonal branch addition and a decrease in the numbers of mature synapses [43], an observation consistent with our TrkB-Fc results. In particular, because those studies were performed by confocal imaging using fluorescence excitation light that would be visible to the tadpole, the conditions during imaging in those experiments would most likely resemble our synchronous stimulation paradigm.

In our daily imaging experiments of contralaterally projecting axons, knockdown of TrkB and p75$^{NTR}$ gave opposite effects on axon arbor growth and branch formation after several days (**Figs 3 and 4**). This observation stands in contrast to our data on rapid branch dynamics in ipsi axons where both p75-KO and TrkB-KO manipulations appear to have decreased branching in response to asynchronous visual stimulation. One must consider, however, that

unlike ectopic ipsi axons, the contralateral RGC axons exhibit coarse topographic order within the optic tectum and thus, on average exhibit a pattern of firing that is relatively correlated with that of their neighbors [7,9–14,54]. It is therefore likely that correlated neural activity dominates among contra axons, obscuring the effects of Stentian plasticity for all but the most imprecisely targeted stray axons over days. The short-term dynamic imaging of ipsi axons, which allowed us to systematically control the degree of correlation in RGC firing, unmasked roles for presynaptic p75[NTR] and TrkB in contributing to Stentian structural plasticity through an increase in the rate of new branch addition (**Fig 1D and 1F**).

Furthermore, there are numerous examples of the same manipulation exerting different outcomes on short-term versus longer-term morphological changes of RGC axons in the optic tectum. For example, blockade of NMDAR [9,10], as well as TrkB.T1 overexpression [43] both profoundly alter rapid branch addition and loss rates without manifesting significant changes in arbor size at 24 h. Although it is tempting to infer that long-term arbor remodeling is the direct cumulative consequence of the rapid dynamic branch behaviors observed over minutes, it may be more appropriate to think of rapid dynamics as reflecting the activation of underlying plasticity mechanisms rather than the outcome of the refinement process itself. In the current study and in our previous work, we observed that asynchronous stimulation led to enhanced ipsi axon branch dynamics within minutes, but that arbors that had previously been exposed to Hebbian stabilizing stimuli were more refractory to exploratory growth [7]. Thus, the dynamic mechanisms revealed by our unique experimental design would be expected to produce a range of intermediate outcomes depending on the animal's recent and ongoing sensory experience, with more consistent changes emerging after several days of natural visual experience (**Figs 3F, 3G** and **4**).

This raises a potential caveat in our experimental design that should be considered. Although we believe the ipsi RGC axons to be the result of a serendipitous axon guidance error and thus not a unique class of RGC, an important difference between ipsi axons and the typical contralaterally projecting axons is their past history of correlation with neighboring inputs. We cannot exclude the possibility that ipsi axons, having developed for many days under atypical conditions of constant asynchronous visual experience, could have a very different molecular signaling or transcriptional profile compared to contralaterally projecting axons that experience primarily synchronous activity. It is at least reassuring that ipsi axons do not appear to be more susceptible than other RGCs to undergo apoptosis ($p = 0.375$, log-rank test for survival, $n = 6$ ipsi, 15 contra). In addition, although neurotrophin signaling has been strongly implicated in RGC survival and cell death in disease and development, we also found that p75-MO and TrkB-MO contra RGCs had survival profiles that were indistinguishable from control axons over 4 days of imaging (**S5 Fig**), indicating that at the stages when we performed our imaging experiments, these RGCs did not experience excess apoptotic pressure.

## Local action of BDNF signaling in Hebbian branch stabilization

Our analyses on the spatial distributions of addition and elimination of axonal branches revealed that correlated firing led to a decrease in the mean distances between elimination events compared to what occurs in darkness, specifically a loss of the longest pair distances, suggesting that the arbor is subjected to local influences on branch elimination (**Fig 2D–2F**). In contrast, the mean distance between branch addition events stayed similar regardless of stimulation. These data are consistent with previous observations in tadpoles with binocularly innervated optic tecta showing that axonal branches are added uniformly across the arbor, but branch loss tends to occur in the territory where inputs from the opposite eye dominate [11]. A decrease in the mean pairwise distance of elimination events could occur through localized

action of axonal branch elimination signals or through locally diffusible signals that protect axonal branches from being eliminated in parts of the arbor where the firing of the axon is better correlated with activity in its local postsynaptic partners. We found that disrupting BDNF signaling in the retinotectal system resulted in increased rates of activity-dependent branch elimination (**Fig 1E and 1G**) and these elimination events occurred more ubiquitously throughout the arbor (**Fig 2E**). Together, our data (**Figs 1G** and **2E**) suggest that release of BDNF at sites where inputs are mutually correlated confers local axonal branch stabilization, such that blocking BDNF signaling increases the relative frequency of elimination events, even under conditions of correlated neural activity. These findings together with previous reports that BDNF increases the synaptic clustering on RGC arbors [35] suggest a model in which local BDNF signaling results in localized synaptic and structural stabilization (**Fig 5**).

## Molecular mechanisms of axonal "competition" in the optic tectum

It has been shown that activity-dependent release of pro-neurotrophin acts on axonal p75$^{NTR}$ to induce axonal pruning of outcompeted inputs in the peripheral nervous system, both in sympathetic neurons and at the neuromuscular junction (NMJ) [31,55–57]. Conversely, "winners" of the competition are maintained and strengthened via TrkA-dependent (sympathetic neurons) or TrkB-dependent (NMJ) mechanisms. In the central nervous system, where poly-innervation of postsynaptic partners is predominant, competition appears to occur differently. For example, the "loser" RGC axons do not undergo apoptosis or retraction in response to competition, but rather undergo synaptic weakening and exhibit more dynamic growth and branching ([6–8]). Furthermore, our data for the retinotectal projection provide evidence that activity per se is not sufficient to stabilize the "winning" axon, but rather correlated activity inducing postsynaptic NMDAR activation appears to be required. Nevertheless, the involvement of p75$^{NTR}$ on the "punished" axon appears to be a common mechanism. It has been suggested that the distinct morphological consequences on "punished" axons in the peripheral versus central nervous systems may be attributable to different sets of interacting receptors or downstream signaling molecules expressed in these different cells, which, for example, may lead to RhoA activation in the one case versus RhoA inhibition in the other [58]. Our study did not identify the specific ligands that act via TrkB and p75$^{NTR}$ to produce the Stentian increase in branch dynamics, but the potency of extracellularly applied TrkB-Fc implies a released neurotrophin, most likely BDNF or its precursor proBDNF based on expression patterns in tectal development [36,59]. Future studies involving knockdown of p75$^{NTR}$ and its interactors, including Sortilin and Nogo receptor, will be important to further elucidate the precise ligands and downstream signaling in Stentian versus Hebbian plasticity.

Our data suggest a model in which presynaptic signaling through p75$^{NTR}$ and possibly TrkB receptors is required for Stentian exploratory branching, and in which postsynaptic BDNF signaling is necessary to drive Hebbian stabilization that results in suppression of new axon branch addition (**Fig 5**). Furthermore, released BDNF and presynaptic p75$^{NTR}$ mediate Hebbian suppression of branch loss. Opposing pre- and postsynaptic effects on branch initiation could reflect differences in the levels of expression of TrkB and p75$^{NTR}$ and their interactors in dendrites and axons [60–63]. Another possibility is that proBDNF may be released in response to neural activity either from the neighboring RGCs or from the postsynaptic tectal neuron to act on p75$^{NTR}$ (**Fig 5B**), but can be further converted to mBDNF through NMDAR-dependent activation or release of proconvertases such as tissue plasminogen activator (tPA) or matrix metaloproteinase-9 (MMP-9), leading to preferential activation of TrkB during synchronous stimulation (**Fig 5C**) [64–66]. tPA mediates synaptic strengthening in the retinotectal projection in response to visual conditioning [36]. MMP-9 has been implicated as a

molecular switch (via proBDNF-to-mBDNF cleavage) to convert axon pruning to stabilization in the *Xenopus* NMJ and synaptic weakening to strengthening in rodents [30,31,67]. In the *Xenopus* optic tectum, MMP-9 is necessary for the visual experience driven increase in tectal neuron dendritic growth [68]. It is also plausible that the mode of BDNF release varies during asynchronous and synchronous stimulation, in line with reports proposing that tonic versus acute release of BDNF may differentially regulate the cell surface level of TrkB and thus distinctly affect neurite outgrowth [69–71].

In summary, our experiments have used a unique visual stimulation protocol and targeted knockdown of BDNF receptors to reveal that distinct receptor classes and sites of BDNF signaling underlie Stentian and Hebbian structural plasticity during development (**Fig 5**). Based on our data and the literature, we propose that presynaptic p75$^{NTR}$ and possibly TrkB signaling promote Stentian exploratory growth of retinotectal axons in response to asynchronous activity (probably via proBDNF release). Furthermore, correlated firing results in NMDAR-dependent release (or pro-conversion) of mBDNF that induces Hebbian synaptic strengthening in postsynaptic tectal neurons and delivery of retrograde stabilization signals that suppress new axonal branch addition and locally restrict axonal branch elimination (**Fig 5A and 5C**).

## Materials and methods

### Accessibility

Any reagents generated for this study will be made available upon request to the lead contact. Alternatively, the requestor will be directed to a public repository tasked with distributing the reagent.

Raw axon reconstructions from which the analyses were performed are available at https://figshare.com/projects/BDNF_signaling_in_correlation-dependent_structural_plasticity_in_the_developing_visual_system/160870.

Custom MATLAB software (CANDLE denoising) used in this study is available for free download at https://sites.google.com/site/pierrickcoupe/softwares/denoising/multiphoton-image-filtering.

Custom arbor morphology analysis code is available at https://github.com/ekutsarova/Dynamic-arbor-morphology-analysis.

### *Xenopus laevis* tadpoles

The experiments described in this study were approved by the Animal Care Committee at the Montreal Neurological Institute (AUP 7728) and are in accordance with the guidelines of the Canadian Council on Animal Care. Animals of both sexes were used for the experiments. At the stages of interest for our study, the sexes cannot be distinguished anatomically. Albino *Xenopus laevis* tadpoles were generated by induced mating of a sexually mature female frog, injected with pregnant mare serum gonadotropin (50 IU) 3 days before mating and with 400 IU human chorionic gonadotropin (HCG) immediately before mating, and a sexually mature male frog injected with 150 IU HCG up to 8 h before mating. The produced fertilized eggs were reared in 0.1X modified Barth's saline with HEPES (MBSH).

### Morpholino oligonucleotides

*Xenopus* p75$^{NTR}$ morpholino sequence: 5′-CCA TGC TGA TCC TAG AAA GCT GAT G-3′, referred to as p75-MO, *Xenopus* TrkB morpholino sequence: 5′-CCA CTG GAT CCC CCC TAG AAT GGA G-3′ [31,49], referred to as TrkB-MO, and standard control: 5′-CCT CTT ACC TCA GTT ACA ATT TAT A 3′ (Gene Tools, LLC), referred to as Ctrl-MO, were generated and tagged with lissamine on the 3′-end (Gene Tools, LLC).

## TrkB-MO validation experiments

Albino *Xenopus laevis* embryos were microinjected at the 2-cell stage with 18 ng of Morpholino (TrkB-MO or Ctrl-MO) in each blastomere, using an automatic pressure microinjector (Harvard Apparatus) with micropipettes pulled from glass (6.66 μL, Drummond Scientific), using a Flaming Brown Micropipette Puller (P-97, Sutter Instruments CO.). The morphant tadpoles were screened for lissamine fluorescence at st. 46 [72]. Aside from 1 phospho-Trk antibody (Cell Signalling, 9141, AB_2298805), the antibodies against TrkB that we tested showed lack of specificity to the *Xenopus laevis* TrkB. We therefore validated the TrkB-MO by measuring the level of phospho-TrkB in response to BDNF treatment between animals injected with Ctrl-MO and TrkB-MO using the phospho-Trk antibody.

Stage 46 *Xenopus laevis* tadpoles injected at the 2-cell stage with TrkB-MO or Ctrl-MO and positive for lissamine fluorescence were injected intraventricularly with 100 ng/μL BDNF (B-250, Alomone Labs) using the same setup described for electroporation-related DNA injections. After 1 h, brains were dissected and extracted in NP-40 extraction buffer (10 mM HEPES/NaOH (pH 7.4), 150 mM NaCl, 2 mM EDTA (pH 8.0), 1% NP40) with Halt protease/phosphatase inhibitor cocktail (Thermo Scientific, P178442). Brain extracts were then homogenized using 10 s continuous 10% Amplitude on a Branson 250 Sonifier (Branson Ultrasonics) equipped with a microtip. The extracts were centrifuged at 13,000 rpm at 4°C for 10 min using a Biofuge 13 centrifuge (Heraeus Instruments). Laemmli sample buffer (50 mM Tris/HCl (pH 6.8), 10% Glycerol, 2% SDS, 0.025% Bromphenol Blue, 100 mM DTT) was added to the supernatant and the protein samples were then boiled at 100°C for 5 min for reduction and denaturation of the protein structure. Proteins were separated on SDS-PAGE (8%) transferred to PVDF membranes (Immobilon-P, 0.45 μm, Millipore) using wet transfer in transfer buffer (48 mM Tris base, 39 mM glycine, 0.037% SDS, 20% methanol). Molecular markers of known size (Precision Plus Protein Dual Color Standard, Bio-Rad) were run in parallel with the proteins of interest. Blots were probed with 1:5,000 rabbit anti-p-Trk (Cell Signalling, 9141, AB_2298805), 1:10,000 peroxidase AffiniPure Goat AntiRabbit IgG (H+L; Jackson Immuno Research Laboratories, 111-035-144, AB_2307391) was used for visualization. Blots were blocked using 5% BSA (Fraction V, Fisher Scientific) in TBS-T (20 mM Tris/HCl (pH 7.6), 135 mM NaCl, 0.05% Tween) for p-Trk, and with 5% fat-free milk in TBS-T for β-tubulin. To ensure equal loading, the blots were probed with 1:20,000 rabbit anti-β-tubulin (sc-9104, Santa Cruz Biotechnology, AB_2241191) and 1:20,000 peroxidase AffiniPure Goat AntiRabbit IgG. Immobilon Western Chemiluminescent HRP Substrate (Millipore) was used to visualize the protein bands on the blots, diluted 1:1 in double distilled water. HyBlot Autoradiography films (Denville Scientific) were used to collect images of the protein bands.

## p75-MO validation experiments

Due to a lack of *Xenopus laevis* p75$^{NTR}$-specific antibody, we took an indirect route to validate that the p75-MO knocks down *Xenopus laevis* p75$^{NTR}$. The p75$^{NTR}$ sequence was cloned from cDNA from st. 24 *Xenopus laevis* tadpoles [72]. Total RNA was extracted from st. 24 tadpoles and cDNA was transcribed with Superscript IV (Thermo Fisher, 18090010). The coding sequence (cds) of p75$^{NTR}$ (starting from the ATG, insensitive to the morpholino, MOres-p75$^{NTR}$) and the cds with part of the 5′UTR (nucleotide -29 from ATG, sensitive to the morpholino, p75$^{NTR}$) were PCR amplified using the following forward primers: MOres-p75$^{NTR}$: 5′ ACG TGA ATT CAT GGA AAC CCC TCT G 3′, p75$^{NTR}$: 5′ ACG TGA ATT CCC TCA GCC ATC AGC T 3′. The reverse primer was the same for both sequences, p75$^{NTR}$-reverse: 5′ ACG TAC CGG TAA CAC GGG TGA GGT A 3′. The PCR fragments were cloned in frame with EGFP into pCS2+ vector, resulting in MOres-p75-EGFP and p75-EGFP. The plasmids were

linearized with NotI and mRNA was prepared with the SP6 mMessageMachine kit (Thermo Fisher, AM1340). Animals were injected at the early 2-cell stage with $2 \times 200$ pg of either MOres-p75-EGFP or p75-EGFP followed by injection of $2 \times 18$ ng of p75-MO. Animals were screened for MO (lissamine) and EGFP expression. In the animals expressing p75-EGFP and p75-MO, there was hardly any EGFP visible and lissamine was used to select animals. Whole animals were extracted at st. 26 in NP40 extraction Buffer supplemented with Halt protease/phosphatase inhibitor cocktail (Thermo Scientific, P178442). Samples were incubated with Laemmli Buffer for 20 min at 37°C. Extracts were separated on 8% SDS-PAGE (as described in TrkB-MO validations experiments), transferred to PVDF membrane, and subsequently stained for EGFP (anti-GFP antibody, 1:20,000, abcam ab13970, AB_300798) and goat anti-chicken HRP 1:10,000 (Jackson Immuno Research Laboratories, 103-035-155, AB_233738) and β-tubulin (anti-β-tubulin, 1:20,000, Santa Cruz sc-9104, AB_2241191) and AffiniPure Goat AntiRabbit IgG 1:20,000, as a loading control. Blots were imaged on a BioRad ChemiDoc Imaging System.

### Sparse labeling of contralaterally or ipsilaterally projecting RGC axons

Albino *Xenopus laevis* tadpoles (stages 40 to 42) were anaesthetized by immersion in 0.02% MS-222 (Sigma, A5040) diluted in 0.1X MBSH. The developmental stages were determined using the standard criteria of Nieuwkoop and Faber (1994). EGFP DNA plasmid (pEGFP-N1, Clonetech, 6085–1) and oligonucleotides were pressure-injected into the eye using micropipettes pulled from borosilicate glass with filament (outer diameter 1.0 mm, inner diameter 0.78 mm; Sutter Instruments) on a PC-10 puller (Narishige, Japan) and attached to a custom-built manual pressure injection system. Upon pressure injection of plasmid/morpholino solutions into the eye, current was delivered across 2 custom-made platinum plates placed in parallel. A Grass 9 constant voltage stimulator with a 3 μF capacitor placed in parallel was used to deliver 2 pulses in each direction: 1.6 ms duration, 36 V for contra axons, 2.5 to 3 ms, 36 V for ipsi axons. After electroporation, the animals were placed in a 20°C biological oxygen demand incubator in a 12 h light/12 h dark cycle. For long-term daily imaging of contra RGC axons, RGCs were co-electroporated with MO and plasmid encoding EGFP (4.5 μg/μL: 1.5 μg/μL). To express EGFP in ipsi RGC axons, 2 μg/μL plasmid was pressure-injected. For experiments involving ipsi knockdown of the BDNF receptors solution containing 3 μg/μL: 4 μg/μL MO and EGFP was injected into the eye in order to achieve efficient co-delivery

### Daily imaging of morpholino-loaded/EGFP expressing RGC axons

Animals at st. 45–46 (3 days post-electroporation) containing a single co-labeled RGC axon were selected and imaged over the next 4 days. After anesthetizing the tadpoles by immersion in 0.02% MS-222 (Sigma, T2379) in 0.1x MBSH, they were positioned in custom-made polydimethylsiloxane (PDMS) imaging chambers. Optical section z-series at 1 μm steps of the axons of interest were acquired using an Olympus FV300 confocal microscope converted for multiphoton use, equipped with a 60× LUMPlanFL N 1.0NA water immersion objective. For optimal excitation of the axon-filling EGFP, imaging was carried out at 910 nm once every day for 4 days. To confirm the presence of lissamine-tagged MO in the axons of interest and assure minimal cross-talk between the 2 channels (green: 500 to 550 nm and red: 593 to 668 nm), 3D fluorescence stacks were also acquired at 840 nm where the EGFP excitation is minimal and the lissamine excitation peaks.

### Dynamic imaging of ipsilaterally projecting RGC axons combined with visual stimulation

Animals were given at least 4 days post-electroporation to assure labeling and proper knockdown in ipsi axons and imaging was performed in stage 47–48 tadpoles. The animals were

immobilized by intraperitoneal injection of 2.5 mM tubocurarine hydrochloride pentahydrate (Sigma). In the experiments including acute pharmacological blockade, 50 μg/mL TrkB-Fc (Recombinant Human Chimera, 688-TK, R&D Systems) was injected intraventricularly. The animals were then placed in a custom-built imaging chamber (PDMS), fixed in place with a small drop of 1.8% UltraPure Low Melting Point Agarose (Invitrogen, 16520). The imaging chamber contains 2 channels for fiber optics, one leading to each eye, a perfusion and spillover chambers, as previously described [17]. During the whole imaging session, the tadpoles were perfused with $O_2$-bubbled 0.1x MBSH. Light flashes were delivered separately to each eye using an FG365LEC-Custom optic fiber (ThorLabs) placed in a channel leading to each eye. To generate light flashes Red Rebel 102lm@700mA LEDs (Luxeon Star, Ltd) controlled by STG4002 and MC Stimulus-II software (Multichannel Systems) were used. After stabilizing the tadpoles, the chamber was placed under the two-photon microscope where the animals habituated in darkness for 30 min. Optical section z-series at 1 μm steps of the EGFP-labeled ipsi axons were acquired every 10 min for 1 h in darkness followed by 1.5 h of asynchronous stimulation: 10 ms flashes, alternating between the 2 eyes at a duty cycle of 0.5 Hz for each eye, followed by 1.5 h of synchronous stimulation: simultaneous 10 ms flashes light in the 2 eyes presented at 0.5 Hz. For some experiments involving knockdown of proteins of interest, optical section z-series of EGFP and lissamine co-labeled RGC axons were acquired for 1 h in darkness, 2 h in asynchronous, and 2 h in synchronous stimulation. Imaging was performed at 910 nm allowing optimal excitation of EGFP using a ThorLabs multiphoton microscope equipped with XLUMPlanFL N 1.0 NA 20× WI objective (Olympus), 2 channels with gallium arsenide phosphide photo-multiplier tubes (GaAsP PMTs) filtered to detect green and red emission (525/50 and 630/92) and ThorImage software. One axon from the Control group was electroporated with EGFP only and was imaged for 1 h in darkness, 1.5 h in asynchronous stimulation, and 1.5 h in synchronous stimulation. For some experiments including co-electroporation of EGFP and MO, additional optical section z-series were collected at the end of each imaging session at 840 nm preferentially excite lissamine over EGFP.

## Image analysis

All multiphoton z-series were denoised using CANDLE software implemented in MATLAB (MathWorks) that relies on non-local mean filtering methods [73]. Denoised 3D stacks were then used to reconstruct axonal arbors using "Autopath" and "Autodepth" features in Imaris 6.4.2 (Bitplane) for daily imaging experiments and by manual tracing in Dynamo software [7], implemented in MATLAB (MathWorks), generously provided by Drs. Kaspar Podgorski and Kurt Haas (UBC). For morphometric analysis of daily imaged RGC axonal terminal branch points were extracted from Imaris 6.4.2. PyImarisSWC Xtension, implemented in Python, installed in Imaris 9.5.1, was used to export the Imaris reconstructions as swc-files. The exported node coordinates were converted from pixel to μm. The swc-files were imported using "trees toolbox" [74], which was used to classify axonal segments according to their order derived by the Strahler ordering method, using "strahler_tree" function [75–77]. Axonal segments are defined as axonal structure bordered by 2 branch points or by a branch and a terminal point. The length of axonal segments with Strahler number of one, referred to as terminal segments was extracted and the distribution of terminal segments binned by length was followed over 4 days. The length of all segments with Strahler number>1 was counted toward the total skeleton length. Axonal span volume was calculated after obtaining the convexity of the 3D reconstructed arbors with "convexity_tree" function and then using the value to calculate and feed a shrink factor into "boundary_tree" function of "tree toolbox" as described by [44]. From the dynamic imaging experiments ipsi RGC axon branch additions and eliminations

between 2 consecutive time-series (10 min) were extracted and further normalizations were performed as described in each individual case in the figure legends. In **Fig 1F**, an average of the first hour of each visual stimulation was performed. In **Fig 1G**, an average of the second hour of visual stimulation for Control, p75-MO, and TrkB-MO. For TrkB-Fc and one of the Control cells (for which each visual stimulation duration was 1.5 h), the last 40 min was assessed instead of 1 h. In **Fig 2E and 2F**, calculation of the pairwise distances was performed for 1.5 h of each visual stimulation. In **S3A Fig** and **Fig 1B**, the bin averages were calculated analogously to **Fig 1F** and **1G**, respectively. In **S3C and S3D Fig**, averages were calculated for every hour of the experiment, except for TrkB-Fc and 1 control axon for which each bin of both visual stimulation represents 40 min (due to the fact that the stimulation periods were 1.5 h long). A branch was counted towards any further analysis only if it attained a length of 1.5 μm at some point during the imaging session. In 2 axons, the asynchronous stimulation was 10 min longer, and this extra time point was not used for further analyses. In 2 axons, one time point was missing, and in another one, 2 time points were missing resulting in a longer (20 min) interval between 2 time points. Addition events were defined as the branch points of newly added branches between "t-10 min" and "t." Elimination events were defined as the terminal points of branches lost between "t" and "t+10 min." Rigid body transformation using manual landmarks in Dynamo was applied to align the time points of the reconstructed arbor. The coordinates of all the events throughout a stimulation period—dark, asynchronous, or synchronous, were extracted and the pairwise distances between all the elimination and all the addition events within a stimulation period were calculated. For 1 axon from the TrkB-Fc group, there was only 1 elimination event throughout the dark period, which prevented the calculation of mean elimination event pair distance for the dark period and further normalization. Thus, this cell was excluded from the elimination event analyses. To assess whether the changes in mean distances were explained by changes in axon arbor morphology, a randomization of the events on the axonal arbor was performed. To do that, the reconstructions in Dynamo were used to extract the coordinates of each node and a swc-file was exported for each time point of the imaging session. The reconstruction was then resampled using "resample_tree" from "trees_toolbox" such that the nodes composing the arbor were equidistant at 0.15 μm. The number of observed addition events between "t-10 min" and "t" was randomly distributed on the axonal arbor at "t." The number of observed elimination events between t" and "t+10 min" was randomly assigned to the terminal points of the axonal arbor at "t." Pairwise distances between the simulated addition or elimination events within a stimulation period were calculated and then the mean pairwise distance for stimulation period obtained. This randomization was repeated 100 times and the average simulated mean pairwise distance plotted on **S4A and S4B Fig**. The ratio of the observed (**S4C and S4D Fig**) and simulated (**S4A and S4B Fig**) mean pairwise distance was plotted on **Fig 2E and 2F**. The imaging experiments and the axonal reconstructions were performed blind to the experimental design, except for 1 axon in the TrkB-Fc and 1 axon in the Control group.

## Statistical analysis

Aligned rank transform for nonparametric factorial 2-way mixed design model with BDNF manipulation as between-subject and visual stimulation (**Figs 1E, 1F, 2E, 2F, S2, and S4**) or time (**Figs 3E–3G and 4D**) as within-subject factor was performed using ART tool package implemented in R [78]. After obtaining significant interaction, to assess statistical significance of simple main effects, either Kruskal–Wallis or Friedman's test, followed by pairwise post hoc Dunn's tests corrected for multiple comparisons using the Benjamini, Krieger and Yekutieli (BKY) two-stage linear step-up procedure [79] were carried out using GraphPad Prism 9.0.0

and the results of the tests used for each experiment are described for each figure legend and in S1 Table. In some cases (**Figs S3A, S3B** and **4E**) only Kruskal–Wallis test, followed by pairwise post hoc Dunn's tests corrected for multiple comparisons using the BKY two-stage linear step-up procedure were carried out. The axonal reconstructions and their spanning fields in **Fig 4A–4C** were plotted using Plotly.

## Supporting information

**S1 Fig. Related to Fig 1: Validation of p75-MO and TrkB-MO.** (A) Schematic of the mRNA constructs encoding p75$^{NTR}$-EGFP fusion protein. Binding of p75-MO to the $p75^{NTR}$-5′UTR-containing mRNA (p75-EGFP mRNA; left) impedes mRNA translation, whereas the inability of p75-MO to bind the construct lacking the $p75^{NTR}$ 5′UTR (MO-resistant p75-EGFP mRNA; right) spares mRNA translation. (B) Schematic of mRNA construct and MO injection at 2-cell stage, raising injected embryos to stage 26 and preparation of whole-animal homogenates. (C) Western blot analysis of homogenates derived from animals injected with combination of p75-EGFP mRNA or MO-resistant p75-EGFP mRNA and p75-MO probed for EGFP and β-tubulin. (D) Schematic of MO injection at 2-cell stage, followed by intraventricular injection of BDNF in morphant tadpoles at stage 46, 1 h before brain homogenate preparation. (E) Western blot analysis of brain homogenates probed for phospho-Trk (p-Trk) and β-tubulin. Approximate size (kDa) is shown next to the bands. The original western blot images used to generate S1 Fig can be found in S1 Raw Images.
(TIF)

**S2 Fig. Related to Fig 1: Replication of the effects in ipsilaterally projecting RGC axon branch dynamics observed by Munz and colleagues, 2014 [7].** Branch (A) additions and (B) eliminations from the control group in the current study (black) plotted together with the control DAS group in Munz and colleagues, 2014 (Fig 4C and 4D), with the statistical significance shown as discovered in that study (gray). One-way nonparametric Friedman test was followed by multiple comparisons—uncorrected Dunn's test (*$p \leq 0.05$, **$p \leq 0.01$), as the control group from the current study (black) is a replication study rather than a new discovery. Statistical significance as discovered by Munz and colleagues, 2014 (Fig 4C and 4D), # $p \leq 0.05$, ## $p \leq 0.01$, ### $p \leq 0.001$, #### $p \leq 0.0001$. Data represent mean ± SEM. Control from current study ($n$ = 12), control from Munz and colleagues, 2014; DAS ($n$ = 9). The data used to generate S2A and S2B Fig can be found in S7 Data.
(TIF)

**S3 Fig. Related to Fig 1: Blocking distinct components of BDNF signaling affects the time course of branch dynamics changes during asynchronous and synchronous stimulation in ipsilaterally projecting RGC axons.** Branch (A) addition in the first hour of visual stimulation and (B) loss in the second half of visual stimulation are plotted as ratios of asynchronous to synchronous stimulation. One-way nonparametric Kruskal–Wallis for branches (A) added ($p$ = 0.0556) and lost ($p$ = 0.0346) test was followed up by pairwise post hoc tests corrected for multiple comparisons (BKY two-stage linear step-up procedure): *$p \leq 0.05$. Branch (C) addition and (D) loss shown as average over 1 h for the MO-containing groups and 40 min average for TrkB-Fc. Data represent mean + SEM. Control ($n$ = 12), TrkB-Fc ($n$ = 7), p75-MO ($n$ = 7), TrkB-MO ($n$ = 7). The data used to generate S3A and S3B Fig can be found in S8 Data, S3C and S3D Fig in S9 Data.
(TIF)

**S4 Fig. Related to Fig 2: Monte Carlo simulation of addition and elimination event pair distances in ipsilaterally projecting RGC axons.** Mean pair (C) elimination and (D) addition

and mean simulated (A) elimination and (B) addition event pair distances in DAS (1 h, 1.5 h, 1.5 h) derived by redistribution of events at random within the reconstructed arbor. Significant interaction in the two-way mixed design model: (A–C) $p < 0.001$; (D) $p = 0.00866$. Pairwise post hoc tests corrected for multiple comparisons (BKY two-stage linear step-up procedure): $^{*}p \leq 0.05$, $^{**}p \leq 0.01$. Data represent mean + SEM. (A–D) Control ($n = 12$), p75-MO ($n = 7$), TrkB-MO ($n = 7$), (A and C) TrkB-Fc ($n = 6$), (B and D) TrkB-Fc ($n = 7$). The data used to generate S4 Fig can be found in S10 Data.
(TIF)

**S5 Fig. Related to Fig 3: Survival analysis of p75-MO and TrkB-MO contralaterally projecting RGCs.** Survival proportions of contralaterally projecting RGC observed by daily 2-photon imaging in the optic tectum over 4 days. Day *0* corresponds to stage 45–46. CTRL-MO ($n = 29$), p75-MO ($n = 19$), TrkB-MO ($n = 21$). Log-rank test; $\chi^2 = 0.3332$; $p > 0.1$. The data used to generate S5 Fig can be found in S11 Data.
(TIF)

**S1 Data. Spreadsheet containing data used to generate Fig 1D and 1E.**
(CSV)

**S2 Data. Spreadsheet containing data used to generate Fig 1F and 1G.**
(CSV)

**S3 Data. Spreadsheet containing data used to generate Fig 2E and 2F.**
(CSV)

**S4 Data. Spreadsheet containing data used to generate Fig 3E–3G.**
(CSV)

**S5 Data. Spreadsheet containing data used to generate Fig 4D.**
(CSV)

**S6 Data. Spreadsheet containing data used to generate Fig 4E.**
(CSV)

**S7 Data. Spreadsheet containing data used to generate S2 Fig.**
(CSV)

**S8 Data. Spreadsheet containing data used to generate S3A and S3B Fig.**
(CSV)

**S9 Data. Spreadsheet containing data used to generate S3C and S3D Fig.**
(CSV)

**S10 Data. Spreadsheet containing data used to generate S4 Fig.**
(CSV)

**S11 Data. Spreadsheet containing data used to generate S5 Fig.**
(XLSX)

**S1 Table. Table of statistical tests and scores used in this manuscript organized by figure.** The data used to generate S1 Table can be found in S2–S8 and S10 Data files.
(DOCX)

**S1 Raw Images. Uncropped western blots used to generate S1 Fig.** Page 1: The 4 rightmost lanes of this blot, stained for EGFP and tubulin (loading control) were used to generate S1C Fig. Page 2: The 2 rightmost lanes of this blot probed for p-Trk were used to generate S1E Fig.

Page 3: The same blot as on Page 2, stripped and probed for tubulin as a loading control for S1E Fig.
(PDF)

## Acknowledgments

We thank Kurt Haas (UBC) and Kaspar Podgorski (U of T) for Dynamo software, Kelly Sakaki (UBC) for ideas on the imaging setup, Peter Donhauser (ESI) for discussions on image and statistical analysis, Tasnia Rahman (McGill) and Philip Kesner (McGill) for experimental blinding.

## Author Contributions

**Conceptualization:** Elena Kutsarova, Martin Munz, Alex Wang, Edward S. Ruthazer.

**Data curation:** Elena Kutsarova.

**Formal analysis:** Elena Kutsarova, Martin Munz, Alex Wang, Yuan Yuan Zhang, Olesia M. Bilash, Edward S. Ruthazer.

**Funding acquisition:** Edward S. Ruthazer.

**Investigation:** Elena Kutsarova, Anne Schohl, Martin Munz, Alex Wang, Yuan Yuan Zhang, Olesia M. Bilash.

**Methodology:** Elena Kutsarova, Anne Schohl, Martin Munz, Olesia M. Bilash.

**Project administration:** Elena Kutsarova, Edward S. Ruthazer.

**Resources:** Elena Kutsarova, Anne Schohl, Martin Munz, Olesia M. Bilash, Edward S. Ruthazer.

**Software:** Elena Kutsarova.

**Supervision:** Edward S. Ruthazer.

**Validation:** Anne Schohl, Yuan Yuan Zhang.

**Visualization:** Elena Kutsarova, Alex Wang.

**Writing – original draft:** Elena Kutsarova, Edward S. Ruthazer.

**Writing – review & editing:** Anne Schohl, Martin Munz, Alex Wang, Yuan Yuan Zhang, Olesia M. Bilash, Edward S. Ruthazer.

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
