## [Editor Report · Decision Letter 0]

15 Jun 2022

Dear Dr Ruthazer, 

Thank you for submitting your manuscript entitled "BDNF signaling in Hebbian and Stentian structural plasticity in the developing visual system" for consideration as a Research Article by PLOS Biology.

Your manuscript has now been evaluated by the PLOS Biology editorial staff, as well as by an academic editor with relevant expertise. I am writing to let you know that we would like to send your submission out for external peer review.

I also wanted to take a quick moment to say "Hi". know it’s been quite a while, but you may recall me from prior interactions when I was Deputy Editor at Neuron under Katja Brose. I’ve been freelance writing and editing for the past several years, including working as a PLOS Biology freelance editor. A few months ago I started up full time at PLOS Biology, working as the Neuroscience Senior Editor and Section Manager covering neurosciences, physiology, chronobiology, stem cell biology and translational medicine. I look forward to continued interactions with you going forward on this submission.

Before we can send your manuscript to reviewers, we need you to complete your submission by providing the metadata that is required for full assessment. To this end, please login to Editorial Manager where you will find the paper in the 'Submissions Needing Revisions' folder on your homepage. Please click 'Revise Submission' from the Action Links and complete all additional questions in the submission questionnaire.

Once your full submission is complete, your paper will undergo a series of checks in preparation for peer review. After your manuscript has passed the checks it will be sent out for review. To provide the metadata for your submission, please Login to Editorial Manager (https://www.editorialmanager.com/pbiology) within two working days, i.e. by Jun 17 2022 11:59PM.

Kind regards,

Kris

Kris Dickson, Ph.D. (she/her)

Neurosciences Senior Editor/Section Manager

PLOS Biology

kdickson@plos.org

---

## [Decision Letter · Decision Letter 1]

4 Aug 2022

Dear Dr Ruthazer,

Thank you for your patience while your manuscript "BDNF signaling in Hebbian and Stentian structural plasticity in the developing visual system" was peer-reviewed at PLOS Biology. It has now been evaluated by the PLOS Biology editors, an Academic Editor with relevant expertise, and by several independent reviewers. 

In light of the reviews, which you will find at the end of this email, we would like to invite you to revise the work to thoroughly address all of the reviewers' concerns, including conducting new experiments where requested.

Given the extent of revision needed, we cannot make a decision about publication until we have seen the revised manuscript and your response to the reviewers' comments. Your revised manuscript is likely to be sent for further evaluation by all or a subset of the reviewers.

**IMPORTANT - SUBMITTING YOUR REVISION**

*Re-submission Checklist*

*Published Peer Review*

*PLOS Data Policy*

*Blot and Gel Data Policy*

Sincerely,

Kris

Kris Dickson, Ph.D. (she/her)

Neurosciences Senior Editor/Section Manager

PLOS Biology

kdickson@plos.org

REVIEWS:

Reviewer #1: Kutsarova, Ruthazer and colleagues examined the role of BDNF, and its associated receptors TrkB and p75, on Hebbian and Stentian plasticity of ipsilateral retinal ganglion cell axons (RGC) during visual circuit development in albino Xenopus laevis tadpoles. The authors found that the presynaptic receptors, TrkB and p75, promoted Stentian plasticity via RGC axon branch addition. Additionally, BDNF signaling had two significant roles in the Xenopus optic tectum: mediation of Hebbian plasticity via axon stabilization and suppression of local branch elimination. Although the paper describes relevant findings, the significance of the ipsilateral axons in albino Xenopus tadpoles needs to be further elaborated on as the reader is left wondering the exact importance of the findings. The paper also lacks the acknowledgements of significant literature relevant to this study, which can help put the new findings in context with the role for BDNF in Stentian plasticity. The clarity of the presented results could also be improved.

Specific comments: 

There are several issues that the authors need to address in their presentation and discussion of the results, particularly with respect to previous evidence on the role of BDNF signaling on axon branch dynamics.

1. The authors should elaborate on whether ipsilateral RGC axons in albino Xenopus tadpoles at the stage examined do have a functional role in the circuit. This is important since the authors have found that only 20% of the tadpoles do have ipsilaterally projecting RGC axons (Rhaman… & Ruthazer et al, 2020). Are these ipsilateral arbors only found in Albino tadpoles? Is this an error in pathfinding?

2. The authors examined "branch dynamics (addition and elimination of branches), but in many instances the branches measured are very small (1.5 micron) suggesting that these may be transient filopodia. How can one distinguish true branches vs filopodia? Do these filiopodia make active synapses?

3. Treatment to observe effects on ipsilateral RGC axons must have involved transfecting most of the retinal neurons in one eye to then label the single RGC that projects ipsilaterally in that minority of the tadpoles (20%). This then means that the retinal circuit overall in one eye is affected by the manipulation. This is important to discuss or evaluate since both TrkB and p75 signaling have been reported to affect RGC development and differentiation locally within the retina. For example, Hutson and Bothwell 2001 have shown that interfering with p75 signaling affects RGC survival, Lom and Cohen-Cory 1999 and Lom et al 2002 have shown that BDNF affects RGC dendritic differentiation and therefore most likely RGC function locally within the retinal circuit. How can then those significant changes locally within the manipulated retina affect those sole ipsilaterally projecting axons? 

4. The authors need to discuss their results in relation to published studies that have shown that TrkB signaling (dominant negative downregulation in single Xenopus RGCs through specific truncated TrkB expression; Marshak et al 2007) interferes with the dynamic branching of RGC axons (addition and elimination of branches), alters growth cone morphologies, and importantly affects synapse formation (dynamic imaging and EM studies). 

5. Why are changes in branch dynamics are only observed by four days after Morpholino injection (Figure 3), when presumably expression of proteins begins to recover?

6. In general, all graphic data is presented in a manner that is difficult to understand, especially since the bars are noisy and the legends to figures lack detail explanation of what is presented in the graphs. 

7. Lines 247 - 254. Here, the authors need to acknowledge published work that specifically showed a role for BDNF in axon branch stabilization in Xenopus (Hu et al, 2005). Whether those effects could be pre- or postsynaptic was also addressed in another paper studying Xenopus RGC axons published in 2006 (Sanchez et al, Development 2006). 

8. Lines 282-292. The logic in this portion of the discussion somewhat difficult to follow. 

9. Lines 292-300. Here, the authors can also refer to Hu et al, Development 2005 to discuss how their new results are consistent with findings on the role of BDNF and activity on RGC axon branch stabilization in Xenopus.

10. The authors should provide a more detailed explanation for the model shown in Figure 4G.

Reviewer #2: Review of Kutsarova et al., PLoS Biology

"BDNF signaling in Hebbian and Stentian structural plasticity in the developing visual system"

Summary: 

The authors applied a combination of dual-eye optical stimulation, live-cell imaging, morpholino-based receptor knockdowns, and competitive BDNF sequestration to study the role of BDNF signaling in the development of RGC axons in the Xenopus tectum. The key findings are that 1) deletion of p75 and TrkB receptors from RGCs changes axon branching dynamics in response to synchronous/asynchronous RGC stimulation (presynaptic effects), 2) extracellular sequestration of BDNF by application of TrkB-fc protein fragments increases branch addition in axons of synchronized RGCs (postsynaptic effects), and 3) RGC expression of p75, but not TrkB, is required for axonal elaboration. The experiments are rigorous in their design and the data quality is high. This manuscript will be of general interest to researchers investigating mechanisms of neural development and will be of special interest to researchers studying the structure, function, and development of visual circuits as well as mechanisms of Hebbian/Stentian plasticity. There are several concerns that should be addressed prior to publication: 1) the results of the current study do not appear to recapitulate results from a previous study from the same group, 2) the discussion of the different effects of p75 versus TrkB KD on axon development could be more clear, 3) the authors may overstate certain interpretations/conclusions - a more balanced discussion would be helpful, and 4) the presentation/discussion of statistically-significant findings is unclear in several places. 

Overall: 

This reader found it challenging to parse the relative contributions of p75 and TrkB signaling to axon remodeling and a more clear discussion would be helpful. The manuscript will also be improved by including a direct comparison of the current data with previous results from Munz et al. 2014. Analysis of additional biological replicates may help to address issues with variance and trends in the results. 

Major concerns: 

1) The data presented in Figure 1 E-G measuring branch loss do not recapitulate previous results reported from the same group and this raises questions about the interpretation of these experiments on BDNF's role in branch elimination/stabilization. Munz et al. 2014 showed that in a dark/asynchronous/synchronous (DAS) stimulation paradigm, asynchronous stimulation increased branch loss almost two-fold compared with dynamics in the dark. Thereafter, synchronous stimulation did not affect branch loss. These effects were dependent on the stimulus order and DSA stimulation caused increased branch loss following both synchronous and asynchronous stimulation. In the current study, Figure 1 E-G presents the results of DAS stimulation and reports no differences in branch loss across the stimulation paradigm in control ipsiRGCs. In lines 130-134 the authors write: "Knock-down of p75NTR or TrkB in the ipsi RGC axon, as well as broad depletion of BDNF signaling, led to an overall increase in the rates of axonal branch loss in response to visual stimulation compared to control ipsi axons (Figures 1E and 1G). Thus, under normal conditions, BDNF signaling can help stabilize axons against activity-induced axonal branch loss." However, this interpretation is based on the finding that branch loss is essentially unchanged in control ipsiRGCs across the DAS protocol in the current study. This raises several questions that should be addressed prior to publication: 

a. What is the nature of the discrepancy between branch loss levels on control ipsiRGCs measured in the current study and those reported in Munz et al. 2014? If the authors compare their current control data set with the previous data from Munz et al 2014 do they find statistically significant differences in rates of branch addition/loss between control data sets across the two studies? 

b. If the authors perform additional imaging experiments can they replicate the findings of Munz et al 2014 for branch loss and, if so, what would this mean for the findings reported in other experiments with similar replicate numbers to the current control sample data? 

c. The current study uses a dark/asynchronous/synchronous (DAS) stimulation protocol, which according to the Methods appears to be a fixed paradigm for all experiments. Previous work from the group (Munz et al. 2014) shows that branch remodeling dynamics depend on the stimulus order, DAS/DSA versus DSA/DAS. A discussion of how the stimulation history impacts branch dynamics and BDNF signaling would help the reader better understand and contextualize the results of the current study. 

d. The results presented in Figure 2 on branch loss data in control ipsiRGCs are difficult to interpret due to the aforementioned discrepancy between the current study and Munz et al., 2014. A comparison of the current data with those presented in Munz et al. 2014 would be helpful for understanding the lack of any effect on branch loss during the asynchronous stimulation epoch. 

e. The data presented in Figure 1 show significant variance between replicates. The authors should comment on the sources of this variance and how this may impact the interpretation of their results. For example, in panel F the control data show that in four experiments asynchronous stimulation caused a reduction in branch addition while in four other experiments more branches were added. Overall, the cumulative data for Panel F show a statistically-significant increase in branch addition with asynchronous stimulation but the effect is being driven primarily by two animals with 2/2.5x increases in branch addition relative to the dark period. Similar outliers exist in the morpholino data and raise questions about how this variance contributes to the overall conclusions. 

f. Similar to the variance issue described in (e), there are trends in the data that may be clarified by analyzing additional biological replicates. In Figure 4E, it appears that p75-MO axons are not significantly smaller than controls and TrkB-MO axons are not significantly larger than controls, while p75-MO and TrkB-MO are significantly different from one another. Analysis of additional replicates may strengthen the conclusions in this section of the work. 

2) The results from Figure 1 show that TrkB-MO and p75-MO manipulations lead to similar outcomes on branch dynamics of ipsilaterally projecting RGCs. In contrast, the results in Figures 3/4 show that p75-MO leads to reduced complexity of contralaterally projecting RGCs, while TrkB-MO has no effect. What is the interaction/compensation between these two pathways for BDNF signaling? Did the authors perform dual KD of both receptor types in any experiments? In lieu of new experiments, a more thorough discussion would help the reader better understand these results. Currently it is challenging for the reader to reconcile the similar effects shown in Figure 1 with the divergent effects shown in Figure 3/4. In lines 232-238, the authors write: "It is therefore plausible that the level of correlated neural activity among contra axons obscures the effects of Stentian plasticity for all but the most imprecisely targeted stray axons over days. The short-term dynamic imaging of ipsi axons, which allowed us to systematically alter the degree of correlation in RGC firing, unmasked a role for presynaptic TrkB in contributing to Stentian structural plasticity through an increase in the rate of new branch addition (Figures 1D and 1F)." This discussion is confusing. Given that short-term branch dynamics are similar (Figure 1), a clear and thorough discussion of the diverging results between p75 and TrkB in Figure 3/4 is needed. This confusion is captured in lines 303-304 where the authors write: "We propose that presynaptic p75NTR and signaling promote aspects of Stentian exploratory growth…" - exactly which aspects is unclear in the current discussion. 

3) Lines 202-205 the authors may overstate the conclusion: "Our finding that p75NTR promotes the elongation of axonal branches specifically through the extension of the shortest filopodium-like branches (Figure 3G) further clarifies the role of p75NTR in RGC axonal exploratory growth." The current study relies on loss-of-function experiments so the results are more appropriately discussed as "X is necessary/required for Y". More specifically, the authors have not provided direct evidence that p75 promotes axon elongation- this would require overexpression experiments. The authors should take care throughout to not overstate their conclusions. 

Minor concerns:

1) Line 53: "evidence indicates" is vague and this sentence could be more helpful for the reader if it included a mention of the specific data or analytical approach: e.g. "previous in vivo imaging experiments…"

2) Lines 55-56: the transition between the discussion of Hebbian/Stentian plasticity and BDNF function is abrupt. If the authors can expand the text to have a logical segue from activity patterns to BDNF release, this would help maintain logical flow in the writing. The following sentence beginning in line 58: "BDNF can be synthesized and released in a constitutive or activity dependent manner from axonal terminals and dendrites…" helps to capture the idea that activity patterns/levels govern molecular signaling events and should perhaps be moved up in the text to immediately follow the Hebbian/Stentian discussion.

3) Line 103: typo "the entire the" 

4) The images in Figure 1A are confusing. If the MO-lissamine constructs are electroporated into the eye, why are there large fluorescent artifacts in the tectum (magenta signal right panel)? 

5) Lines 156-159: The authors write: "Extracellular depletion of BDNF with TrkB-Fc, or knock157

down of p75NTR or TrkB in ipsi axons, resulted in a redistribution of branch elimination events, no longer favoring the event proximity normally seen with synchronous stimulation (Figure 2E)." Were the results of the p75 KD and TrkB KD statistically significant? If yes, this should be indicated in the Figure and accompanying text. 

6) Line 165: "Axon arbor elaboration over days relies on p75NTR expression in the RGCs" reads somewhat awkwardly. Consider revising this heading title for clarity. Specifically "the RGCs" stood out as awkward and this phrasing appears elsewhere in the text. 

7) Figure 2 requires a scalebar to accompany the reconstructed axons. Figure 1C also requires as scalebar to accompany the reconstructed axons. 

8) The Euclidian measurement approach used to produce the results in Figure 2 precludes any investigation of local interactions along individual branches. Have the authors considered a measurement of branch dynamics using an on-path analysis that relates events by their proximity along individual branches? This would be a more biologically-relevant and informative measurement. 

9) The Figure 3 legend should explicitly state that experiments are performed in contralaterally projecting RGCs. 

10) The statistics in Figure 4 are unclear. The authors write "with TrkB-MO arbors expanding more rapidly over 4 days to occupy a greater volume in the optic tectum compared to p75-MO axons (Figure 4D-E)" (lines 190-192) and later "Indeed, we observed that TrkB knock-down in RGCs led to a notably enlarged axonal arbor span (Figure 4)." (lines 297-298). "Notably larger" is vague- compared to what exactly (p75-MO or controls)? Is this result statistically significant comparing TrkB-MO to controls? Figures 4 D/E should have very clear presentations of the statistical comparisons to help the reader understand the results. 

11) Are the effects of p75-MO shown in Figures 3/4 related to TrkA-mediated axon maintenance? A discussion of Singh et al. 2008 "Developmental axon pruning mediated by BDNF-p75NTR-dependent axon degeneration" would be helpful. 

Reviewer #3: Kutsarova et al. studied activity-dependent axonal refinement of retinal ganglion cells (RGC) in the optic tectum of albino Xenopus laevis tadpoles and analyzed the role of BDNF signaling pathway by suppressing TrkB, and P75NTR expression with morpholino oligonucleotide or quenching secreted BDNF with TrkB-Fc antibody. First, the authors used multiphoton live imaging combined with visual stimulation. They observed the few ipsilaterally projecting RGC axonal branch addition during uncorrelated visual activation is dependent on both p75 and TrkB but not BDNF. Then they found that BDNF suppresses branch elimination during correlated firing.

While the quantitative measures of axonal growth and retraction are attractive, the results are confusing and disjointed as ipsilateral axons are analyzed in Figures 1 and 2, but contralateral axons are used in Figure 3 (and Figure 4 is unclear). The distinct roles of TrkB and P75NTR are not well characterized. Furthermore, Discussion is full of conjectures and retrograde signaling for Stentian plasticity; the regulation of proBDNF and mBDNF and the local concentration of BDNF are discussed, but no data are presented. Also, the definition of Hebbian plasticity in this context is unclear whether the authors meant to argue LTP- or LTD-like phenomena. 

* Authors mention that "proBDNF signaling through presynaptic p75NTR is required for axonal retraction, whereas mBDNF signaling through presynaptic TrkB leads to axonal stabilization" (lines 67-68). If proBDNF-P75 is necessary for axonal retraction, why do the results in Figure 1 show that p75-MO prevents branch addition? How is it possible to reconcile? Does Stentian plasticity depend on both TrkB and P75NTR? 

* "The Hebbian suppression of new branch additions…presumably the postsynaptic terminal, to induce Hebbian retrograde signals that inhibit the formation of new axonal branches" (line 113-123). The authors present discordant results of TrkB-Fc treated vs. neurotrophin receptor knock-down RGC axon and speculate retrograde signals from the postsynaptic terminal without evidence. How can such a retrograde signal differentiate ipsilateral axons from contralateral axons? Authors should provide supporting data.

* Data in Figure 3 is obtained from contralateral RGC axons (line 169) over four days. How does this relate to the data in Figures 1 and 2, which examined ipsilateral axons? Did the authors examine contra axonal refinement with the DAS system in Figures 1 and 2 and analyze ipsilateral axonal morphology over four days in Figure 3?

* It is unclear whether Figure 4 analyzed ipsi- or contra-lateral axons. Another confusing aspect of the manuscript is that the authors go back and forth between ipsilateral and contralateral axon data and Stentian plasticity and Hebbian plasticity. 

* In lines 208-211, "From experiments where we electroporated p75-MO exclusively in the ipsi eye, we can conclude that p75NTR specifically on the asynchronously firing RGC axon mediates the response to Stentian signals that promote branch addition and extension." TrkB MO also has a similar effect, so it is still not conclusive, and more questions remain, such as the question of proBDNF vs. mBDNF or the identity of the putative retrograde signal.

* In Figure 1C, how is the number of transient axonal terminals counted in the graph?

* In Figure 1 and G, the distribution of plots looks like there are two populations of response; one population that go up with asynchronous stimulation and then go down with synchronous stimulation, and the other population that declines over asynchronous and synchronous stimuli. In the former population, a few outliers appear to dictate the increase during asynchronous stimulation.

* In Figure 1E, why does each curve not start from 1.0?

---

## [Decision Letter · Decision Letter 2]

16 Feb 2023

Dear Dr Ruthazer,

Thank you for your patience while we considered your revised manuscript "BDNF signaling in Hebbian and Stentian structural plasticity in the developing visual system" for publication as a Research Article at PLOS Biology. This revised version of your manuscript has been evaluated by the PLOS Biology editors, the Academic Editor and the original reviewers.

Based on the reviews and discussion with our Academic Editor, we are likely to accept this manuscript for publication, provided you satisfactorily address the data and other policy-related requests at the bottom of this email, as well as the few suggestions raised by Reviewer 2. Please make sure to address all of these issues fully as failure to do so will delay processing of your submission.

When revising your study for resubmission, we'd also suggest a slight title change to make the work more accessible for our broad biology audience. Please consider something like: BDNF signaling acts via distinct structural plasticity mechanisms at pre- and post-synapses to refine the visual circuitry

We expect to receive your revised manuscript within two weeks. 

*Published Peer Review History*

*Press*

Sincerely,

Kris

Kris Dickson, Ph.D., (she/her)

Neurosciences Senior Editor/Section Manager,

kdickson@plos.org,

PLOS Biology

DATA POLICY:

Note that we do not require all raw data. Rather, we ask that all individual quantitative observations that underlie the data summarized in the figures and results of your paper be made available as this information is essential for readers to assess the analyses and to reproduce it. Thank you for providing Supplemental Table 1 and the Supporting Information (Spreadsheets.zip) files for the majority of your data.

1) Please also include summary data for the Supplemental Fig5 graph.

2) Please ensure that figure legends in your manuscript include information on where the underlying data can be found (e.g. “The underlying data supporting Fig X, panel Y can be found in file Z.”).

3) Please also ensure that your supplemental data files have legends, and that these also direct readers to where the summary data can be found.

Please ensure that your Data Statement in the submission system is also updated to accurately describe where your data can be found.

We require the original, uncropped and minimally adjusted images supporting all blot and gel results reported in an article's figures or Supporting Information files. We will require these files before a manuscript can be accepted so please prepare and upload them now. Please carefully read our guidelines for how to prepare and upload this data: https://journals.plos.org/plosbiology/s/figures#loc-blot-and-gel-reporting-requirements

Please provide the full (uncropped) raw file the gel images in Supplemental Fig1 C and E.

DATA NOT SHOWN?

- Please note that per journal policy, we do not allow the mention of "data not shown", "personal communication", "manuscript in preparation" or other references to data that is not publicly available or contained within this manuscript. Please check over your submission carefully and either remove mention of any such data or provide figures presenting the results and the data underlying the figure(s).

Reviewer remarks:

Do you want your identity to be public for this peer review?

Reviewer #1: No

Reviewer #2: No

Reviewer #3: No

***Reviewer #1: The authors have addressed most of my concerns. Clarity of presentation could still be improved. 

***Reviewer #2: The authors have made substantial changes to the manuscript to improve the data presentation and associated discussion. My concerns regarding variability and relationships to previously published results have been addressed. 

A few minor suggestions for revision:

1) A dashed line is used to show normalized control values in several figures, but these lines are missing in Figure panels 3E/F. 

2) In Figure 1G the Y axis is broken and perhaps the relevant bar plots or error bars should also be broken for clarity. 

3) Lines 371-372: "In our daily imaging experiments..." - for clarity this sentence should mention contralateral axons. e.g. "In our daily imaging experiments of contralateral axons, ..."

***Reviewer #3: The authors addressed major points raised by this reviewer and have improved clarity of the manuscript.

---

## [Editor Report · Decision Letter 3]

8 Mar 2023

Dear Dr Ruthazer,

Thank you for the submission of your revised Research Article "BDNF signaling in correlation-dependent structural plasticity in the developing visual system" for publication in PLOS Biology. In the absence of Kris Dickson form the office, I have taken over handling of your manuscript to avoid unnecessary loss of time. On behalf of my colleagues and the Academic Editor, Matthew Dalva, I am pleased to say that we can in principle accept your manuscript for publication, provided you address any remaining formatting and reporting issues. These will be detailed in an email you should receive within 2-3 business days from our colleagues in the journal operations team; no action is required from you until then. Please note that we will not be able to formally accept your manuscript and schedule it for publication until you have completed any requested changes.

PRESS

With best wishes, 

Nonia

Nonia Pariente, PhD, 

Editor-in-Chief

PLOS Biology

npariente@plos.org